# The presynaptic ribbon maintains vesicle populations at the hair cell afferent fiber synapse

Lars Becker[1], Michael E Schnee[1], Mamiko Niwa[1†], Willy Sun[2], Stephan Maxeiner[3‡], Sara Talaei[1], Bechara Kachar[2], Mark A Rutherford[4], Anthony J Ricci[1,3]*

[1]Department of Otolaryngology, Stanford University, Stanford, United States;
[2]National Institute of Deafness and Communicative Disorders, United States;
[3]Molecular and Cellular Physiology, Stanford University, Stanford, United States;
[4]Department of Otolaryngology, Washington University, St. Louis, United States

*For correspondence:
aricci@stanford.edu

Present address: †Department of Otolaryngology, Johns Hopkins University, Baltimore, United States; ‡Department of Neuroanatomy, Institute for Anatomy and Cell Biology, Medical School Saarland University, Saarbrücken, Germany

Competing interests: The authors declare that no competing interests exist.

**Abstract** The ribbon is the structural hallmark of cochlear inner hair cell (IHC) afferent synapses, yet its role in information transfer to spiral ganglion neurons (SGNs) remains unclear. We investigated the ribbon's contribution to IHC synapse formation and function using KO mice lacking RIBEYE. Despite loss of the entire ribbon structure, synapses retained their spatiotemporal development and KO mice had a mild hearing deficit. IHCs of KO had fewer synaptic vesicles and reduced exocytosis in response to brief depolarization; a high stimulus level rescued exocytosis in KO. SGNs exhibited a lack of sustained excitatory postsynaptic currents (EPSCs). We observed larger postsynaptic glutamate receptor plaques, potentially compensating for the reduced EPSC rate in KO. Surprisingly, large-amplitude EPSCs were maintained in KO, while a small population of low-amplitude slower EPSCs was increased in number. The ribbon facilitates signal transduction at physiological stimulus levels by retaining a larger residency pool of synaptic vesicles.
DOI: https://doi.org/10.7554/eLife.30241.001

## Introduction

The synaptic ribbon is an electron dense structure associated with presynaptic active zones at sensory synapses in the retina, lateral line and inner ear organs. RIBEYE is the major protein associated with the ribbon (*Zanazzi and Matthews, 2009*).It consists of a unique amino-terminal A domain, which forms the ribbon scaffold, and a carboxy-terminal B domain with lysophosphatidic acid acyltransferase activity, which is almost identical to CtBP2, a transcriptional co-repressor (*Schmitz et al., 2000*). The ribbon is anchored to the presynaptic membrane by Bassoon protein in mammalian IHCs (*Khimich et al., 2005*) as well as in retinal photoreceptors and bipolar cells (*Dick et al., 2001*). Although it is comprised predominantly of RIBEYE (*Zenisek et al., 2004*), the ribbon complex may associate with over 30 different synaptic proteins (*Uthaiah and Hudspeth, 2010*; *Kantardzhieva et al., 2012*).

IHCs respond to sound with graded receptor potentials that drive rapid and precise synaptic release, encoding intensity and timing information (*Fuchs, 2005*; *Matthews and Fuchs, 2010*). The ribbon is thought to sustain continuous encoding by tethering a pool of vesicles that replenishes the local pool of docked and primed vesicles upon depletion. Other functions ascribed to the ribbon include: regulating vesicle trafficking from the cytoplasm to the active zone and serving as a vesicle trap to bind freely diffusing vesicles, as a location for vesicle formation, as a physical barrier for $Ca^{2+}$ ensuring a high local $Ca^{2+}$ signal, and as a primary structural component of the synapse anchoring $Ca^{2+}$ channels close to synaptic vesicle release sites (*Parsons et al., 1994*; *Frank et al., 2010*; *Graydon et al., 2011*; *Kantardzhieva et al., 2012*).

Postsynaptic excitatory postsynaptic currents (EPSCs) at ribbon synapses range in amplitude by up to 20-fold. The smallest and largest EPSCs have similar kinetics, which together suggest the existence of a mechanism to synchronize the release of multiple quanta (multivesicular release, MVR; [*Glowatzki and Fuchs, 2002*; *Li et al., 2009*; *Schnee et al., 2013*; *Rudolph et al., 2015*]). By holding vesicles and voltage-gated $Ca^{2+}$ channels at the active zone, the ribbon may support MVR, occurring through various mechanisms including: (i) coordinated fusion of multiple single vesicles, (ii) release of large pre-fused vesicles, (iii) rapid sequential fusion in which the first fusion event triggers fusion of additional vesicles, or (iv) some combination of these mechanisms (*Matthews and Fuchs, 2010*).

Previous attempts to study ribbon function used genetic disruption of the anchoring protein Bassoon or acute destruction of the ribbon by fluorophore-assisted light inactivation (FALI). These manipulations resulted in smaller vesicle pool size, calcium channel mislocalization, increased auditory thresholds, asynchrony in neural responses and reductions in both tonic and phasic responses (*Snellman et al., 2011*; *Jing et al., 2013*). However, neither of those manipulations were selective for the ribbon and so might be manifestations of accessory protein denaturing.

Recently, manipulation of the gene encoding CtBP2/RIBEYE allowed for deletion of RIBEYE by removing the A-domain of *ctbp2* to eliminate ribbons in mice, enabling a more precise examination of ribbon function. This mouse will be referred to as KO and the littermate controls as WT throughout the manuscript. Elimination of ribbons in the retina reduced phasic and tonic transmission and rendered release sites sensitive to the slow calcium buffer EGTA (*Maxeiner et al., 2016*). Here we present analysis of cochlear synaptic anatomy, hair cell and afferent fiber physiology and hearing function in the RIBEYE knockout mouse. Results are distinct from those in the retina and from a RIBEYE deletion in zebrafish (*Lv et al., 2016*), suggesting a more subtle function for the ribbon than previously assumed.

## Results

### IHC-sSpiral ganglia neuron (SGN) synapses are formed and maintained despite absence of the ribbon

The inner ear structure of the KO was assessed first for the presence of synaptic ribbons and synapses (*Maxeiner et al., 2016*). Validating loss of hair cell ribbons was accomplished using immunohistochemistry for the RIBEYE B-domain (CtBP2). Anti-CtBP2 labelled the ribbons in wildtype (WT) but not in the KO animals (postnatal day 21, P21)(*Figure 1A*). Nuclear CtBP2 was still observed in both WT and KO (not shown) because the KO is specific to the RIBEYE A-domain. The nuclear staining served as a usful control for antibody function.

Using antibodies to Homer for postsynaptic labeling and Bassoon for presynaptic labeling (*Figure 1B,C*), we quantified the number of puncta and determined whether puncta were juxtaposed as synaptic pairs (*Figure 1I,J*). Bassoon and Homer puncta were similar in number in WT and KO mice (Bassoon — 40 ± 15 in WT, 43 ± 12 in KO, p=0.76; Homer — 17 ± 3 for both WT and KO; WT — n = 20 mice, KO — n = 19 mice, p=0.64). The number of paired Homer and Bassoon puncta (*Figure 1D*) were similarly comparable: 14 ± 2 (n = 20) for WT and 13 ± 2 (n = 19) for KO IHCs (p=0.18), suggesting that synapses were formed and maintained in comparable numbers regardless of the presence of ribbons. The higher number of Bassoon puncta probably reflects its localization at efferent synapses. *Figure 1—figure supplement 1* presents a lower-magnification image that illustrates the multiple roles of Bassoon at the auditory periphery. Bassoon is a major component of the presynaptic elements at the auditory periphery, including the efferent fibers and the hair cell afferent synapses. Thus it is not surprising that paired Bassoon puncta are much fewer in number than total puncta, because Bassoon serves multiple synapses. Similarly, when interpreting Bassoon KO data, it is not surprising that the KO phenotype may be more dramatic than the RIBEYE effect alone due to number of synapses involved and the scaffolding function of Bassoon that leads to structural changes at the synapse.

RIBEYE KO in zebrafish revealed a non-osmiophilic structure resembling a ribbon, which maintained organization of synaptic vesicles in the absence of the RIBEYE protein in hair cells of the lateral line neuromasts (*Lv et al., 2016*). We examined whether such a structure existed in the mammalian cochlea in the IHCs of the KO using transmission electron microscopy (TEM) and

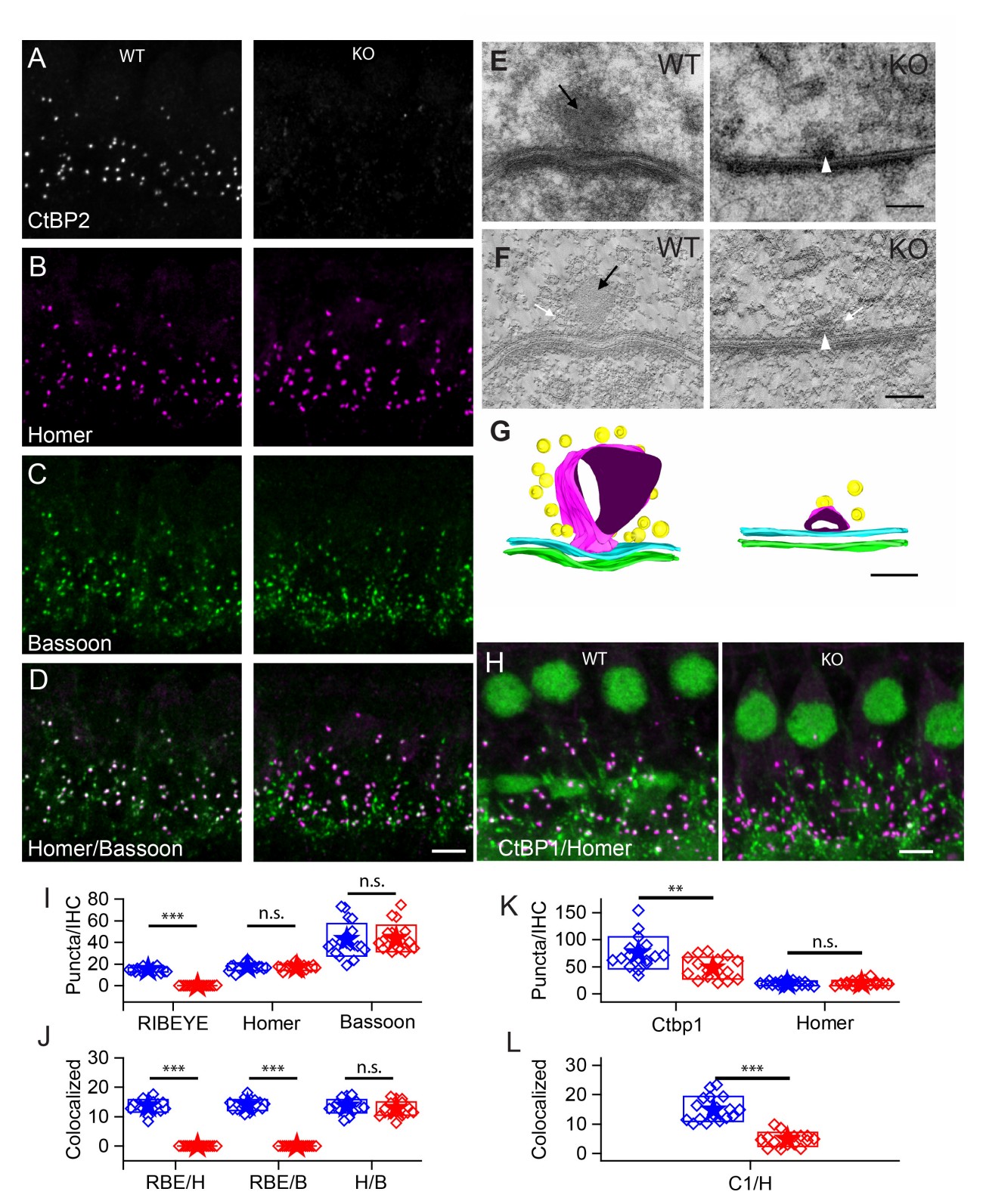

**Figure 1.** Number of synapses stays unaffected in ribbon-less organs of corti. (A–D) Maximum intensity projections of confocal z-sections of inner hair cells (IHCs) in apical cochlea turns of P22 KO and wildtype littermates, immunolabeled for CtBP2/RIBEYE (A), Homer (B), Bassoon (C) and Homer-Bassoon Overlay (D). Virtually no punctated extra-nuclear CtBP2 staining that is colocalized with other synaptic proteins can be seen in KO mice, whereas Bassoon and Homer appear generally unaffected. (E) 100 nm thick transmission electron microscopy (TEM) sections of pre- and postsynaptic

*Figure 1 continued on next page*

*Figure 1 continued*

densities in WT and KO mice at P28 confirm the absence of electron-dense ribbon structures in KO mice. (For E—G) Black arrows point to ribbon structure, white arrows point to synaptic vesicles and white arrowheads point to the remaining density resembling the arciform density. (F) Single slice from a 10-nm thick tomographic section. (G) Reconstruction of complete synapses from data used to create E,F. Presynaptic density in cyan and postsynaptic density in green. Tethered synaptic vesicles modeled in yellow, electron-dense structure attributed to synaptic ribbons in magenta. (H) Confocal maximum intensity projection of IHCs immunolabeled for CtBP1 (green) and Homer (magenta). (I) Quantification of RIBEYE, Homer and Bassoon puncta in the organ of corti normalized to the number of IHCs in analyzed regions of interest, showing complete loss of synaptic RIBEYE staining in KO. The amount of Homer and Bassoon staining is unaffected (p>0.05). WT in blue, n = 20 mice, quantified puncta —RIBEYE = 3796, Bassoon = 10,215, Homer = 4340; KO in red, n = 19 mice, n puncta — RIBEYE = 33, Bassoon = 10,825, Homer = 4201. (J) Paired synaptic puncta per IHC is lost in the absence of RIBEYE staining (p≤0.001) between WT and KO. The number of synapses defined by colocated postsynaptic Homer and presynaptic Bassoon puncta (p=0.18) is unaffected in KO. (K) The number of extra-nuclear CtBP1 puncta is greatly reduced (p=0.003) in KO mice. (WT in blue, n = 9 mice, n puncta — CtBP1 = 19,929, Homer 4940; KO in red, n = 7 mice, n puncta — CtBP1 = 11,960, Homer = 5033). (L) Remaining CtBP1 puncta show significantly reduced (p≤0.001) pairing with postsynaptic Homer staining. Scale bars: A–D, H: 5 µm; E, F: 100 nm. Boxes represent standard deviations of the mean. Significance levels of two-tailed unpaired t-tests: n.s., not significant; *p≤0.05; **p≤0.01; ***p≤0.001.

DOI: https://doi.org/10.7554/eLife.30241.002

The following figure supplement is available for figure 1:

**Figure supplement 1.** Immunohisotchemistry from WT (A) and KO (B) for Bassoon (green), CTBP2 (red) and myosin VIIa (Blue).

DOI: https://doi.org/10.7554/eLife.30241.003

electron-tomography. Representative examples of conventional TEM (*Figure 1E*, 70 nm sections) and tomographic reconstructions (*Figure 1F,G*, 10 nm tomographic sections) are presented. The KO presynaptic synapses consistently showed a small osmiophilic density in contact with the pre-synaptic membrane at the center of the synapse, which resembled the arciform density seen at ribbon synapses in the retina. This density may be the Bassoon-enriched linker between the membrane and the ribbon. Interestingly, this density is not observed in the Bassoon KO mouse (*Frank et al., 2010*). Tomographic reconstructions (*Figure 1G*, right) clearly show the presynaptic density, which is smaller than the ribbon and is present also at ribbon synapses in WT IHCs (*Figure 1G*, left). We found no spatial organization of vesicles near active zones in KO IHCs, suggestive of the 'ghost' ribbons observed in the zebrafish lateral-line hair cells (*Lv et al., 2016*). Thus, in the KO mouse, IHCs lack synaptic ribbons and have no obvious compensatory structures. We surmise that the differences observed in the zebrafish model are a function of the KO being incomplete or there being a compensation mechanism in zebrafish that is not found in mammals.

KO IHCs may upregulate a similar protein to replace the lost RIBEYE. CtBP1 is a good candidate because it is similar to the RIBEYE-B domain, is expressed in hair cells and localizes to the synaptic region (*Kantardzhieva et al., 2013*). Antibody labeling (*Figure 1H,K,L*) demonstrates the presence of CtBP1 in both WT and KO animals, which overlapped with that of Homer in WT animals. A reduction in puncta number was identified in the KO animal due to loss of localization at the afferent synapses (*Figure 1K,L*). Thus, the CtBP1 localization at the synapse requires the presence of RIBEYE, and CtBP1 does not substitute for RIBEYE in the KO animal.

## Synaptic vesicles are normal in size but fewer in number

Vesicle diameters (distance between the outer edges of the bilayer in each vesicle) were measured in the reconstructed tomograms. We measured in both X and Y directions with no differences observed. In the WT, the average vesicle diameter was 33.6 ± 3 nm (n = 21), and in the KO, it was 33.8 ± 3 nm (n = 21).

Synaptic vesicles were conspicuously fewer in number the KO. Vesicles were counted in volumes around the center of the reconstructed synapses (*Figure 1G*), where one boundary was the presynaptic membrane. The volumes had XY dimensions of 453 ± 6 nm for WT (n = 3) and 569 ± 117 nm for KO (n = 11), and Z dimensions of 88 ± 3 and 95 ± 10 nm for WT and KO, respectively. Total vesicle numbers were 20 ± 2 for WT and 7 ± 3 for KO, significantly greater for WT compared with KO synapses (p<0.001, Welch correction for unequal variance). Our TEM data are qualitatively similar to those reported in the companion paper and support the conclusion that ribbons are not present in the KO.

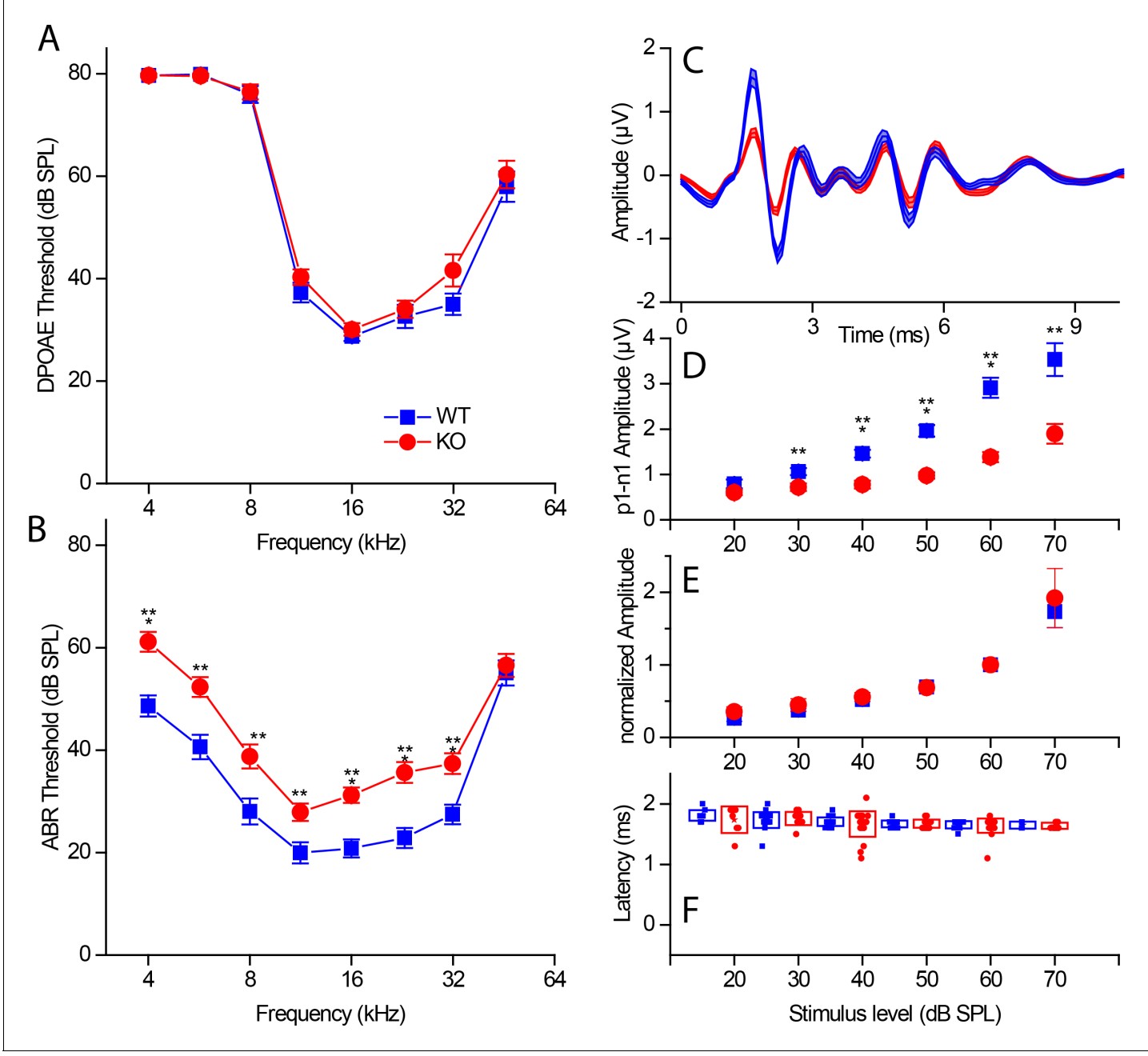

**Figure 2.** Hearing in RIBEYE knockout mice. (A) Distortion products of otoacoustic emissions (DPOAE) show no significant differences in the tested frequency range (4–46 kHz in half-octave increments) between KO (red, n = 30) and WT littermates (blue, n = 22), suggesting normal outer hair cell function. (B) Auditory brainstem response (ABR) measurements in the tested frequency range (4–46 kHz in half-octave increments) reveal a small (≈ 10 dB SPL) but significant threshold shift in KO mice (red, n = 28) over their WT littermates (blue, n = 22) at P21-P22. (C) Mean ABR waveforms to 5 ms tone pips at 23 kHz at 60 dB SPL from KO (red, n = 30) and WT littermates (blue, n = 24) reveal a decrease in amplitude of the first ABR wave in KO. (D) Quantification of peak to peak amplitude (wave I: p1–n1) differences between KO (red) and WT littermates (blue) at 23 kHz were significant over most sound pressure levels (20 dB SPL, p>0.05; 30 and 60 dB SPL, p≤0.01; 40–60 dB SPL, p≤0.001). (E) First peak amplitudes normalized to their respective values at 60 dB SPL. Amplitudes scale with their maximum value in WT and KO. (F) First peak latency (p1) is not significantly altered in KO mice. For wave I amplitude and latency analysis (D–F), the n-number varied between sound pressure levels by a clearly detectable first wave and speaker calibration, n of WT/KO — 20 dB, 11/7; 30 dB, 21/11; 40 dB, 23/22; 50 dB, 23/26; 60 dB, 23/28; 70 dB, 5/6. Boxes represent standard deviations of the mean; whiskers and the shaded line in (C) represent standard errors of the mean. Significance levels of two-tailed unpaired t-tests: n.s., not significant; *p≤0.05; **p≤0.01; ***p≤0.001.
DOI: https://doi.org/10.7554/eLife.30241.004

## Auditory nerve activity is reduced in the absence of synaptic ribbons

Previous investigation of ribbonless synapses in the KO mouse focused on the retina and did not assess function at the systems level (*Maxeiner et al., 2016*). To determine how hearing was affected, we measured auditory brainstem responses (ABRs) and distortion product otoacoustic emissions (DPOAEs). DPOAEs are a direct measure of outer hair cell function and cochlear amplification. DPOAEs are not expected to be affected by the absence of RIBEYE and serve as a control for the developmental acquisition of mature cochlear mechanics. DPOAE threshold measurements were not different across frequencies from 4 to 46 kHz (*Figure 2A*). Data are presented in decibels (dB) as referenced to sound pressure level (SPL) with lower values indicating enhanced sensitivity in the mid-cochlea. Normal DPOAEs indicate that cochlear amplification was not effected mechanically by the loss of outer hair cell ribbons, suggesting that sound-driven activation of IHCs was intact, and that any differences in auditory nerve activity should result from changes at the IHC afferent synapses.

Wave 1 of the ABR is a direct measurement of the synchronized firing of SGNs at sound onset, triggered by synaptic output from cochlear IHCs. The subsequent waves of the ABR reflect the ensuing activation of brainstem nuclei in the ascending auditory pathway (*Zheng et al., 1999*). ABR thresholds were elevated by approximately 10 dB at all probe frequencies in KO mice (*Figure 2B*), except for at 48 kHz, where hearing thresholds are already relatively high. The variance in our control data ranged from 8-12dB depending on frequency; this dictated the sample size needed to detect a 5% change as statistically different. ABR threshold was identified as the stimulus intensity required to evoke a voltage response 5x the root mean square (RMS) noise floor for the measurement. The companion paper found a similar elevation in threshold though it did not reach statistical significance. The difference is probably due to sampling size, where we powered our sample size on the basis of variance found in control measurements. The ABR waveform (*Figure 2C*: 23 kHz, 60 dB) shows a large reduction in wave 1 amplitude with no effect on the subsequent peaks. This was surprising and suggests considerable plasticity in the central processing of data translated from the cochlea periphery. KO mice had smaller wave 1 peak amplitudes, for all sound levels that evoked a response (*Figure 2D*). When wave 1 growth functions were plotted relative to the maximal response amplitude within a genotype, the amplitude versus level function for KO scaled up to overlap with that for WT (*Figure 2E*). Considering wave 1 amplitude as a fraction of the maximal amplitude, the overlap of the normalized wave 1 growth curves suggested that loss of maximum output could produce an elevation of threshold in the ABR measurement, which does not necessarily reflect a sound detection limit or a change in the sensitivity of the auditory system. Indeed, normal ABR amplitudes beyond wave 1 indicate compensation in the cochlear nucleus. Future work should determine whether KO mice have increased behavioral thresholds.

In mice lacking Bassoon function, where IHCs lacked most of their membrane-anchored ribbons, there was an increase in wave 1 latency (*Khimich et al., 2005*) and in first-spike latency in recordings of SGNs in vivo (*Buran et al., 2010*). ABR wave 1 latencies did not differ between WT and KO at any stimulus intensity (*Figure 2F*), suggesting a milder phenotype in the KO as compared to mice with impaired Bassoon function.

As the ABR measurements are averaged responses of 260 stimulus repetitions, a possible explanation for the reduced ABR wave 1 amplitude in KO is that the response reduces over time. If the ribbon is providing a means of maintaining vesicles near the synapse, the concomitant loss of vesicles at ribbonless synapses might reduce the ongoing response to repetitive stimulation. Comparison of ABR amplitudes when the number of repeats was varied from 260 to 1000 showed no difference in amplitudes, nor did repeated measurements meant to evoke depletion. Thus, rundown alone is unlikely to account for the decrease in ABR wave 1.

## Characterization of molecular synaptic components indicates an increase in postsynaptic glutamate receptor distribution

To begin to identify cellular mechanisms that might underlie the measured reduction in ABR wave 1 amplitude, we investigated the distribution of presynaptic $Ca^{2+}$ channels and postsynaptic glutamate receptors as indicators of functional synapses. *Figure 3* presents immunohistochemical localizations of presynaptic ribbons with anti-CtBP2, presynaptic voltage-gated $Ca^{2+}$ channels with anti-$Ca_V$1.3, and postsynaptic AMPA-type glutamate receptors with anti-GluA3 in P35 mice organ of corti. There was no difference in the number of puncta between WT and KO for $Ca_V$1.3 (20 ± 6, n = 9 vs. 18 ± 5,

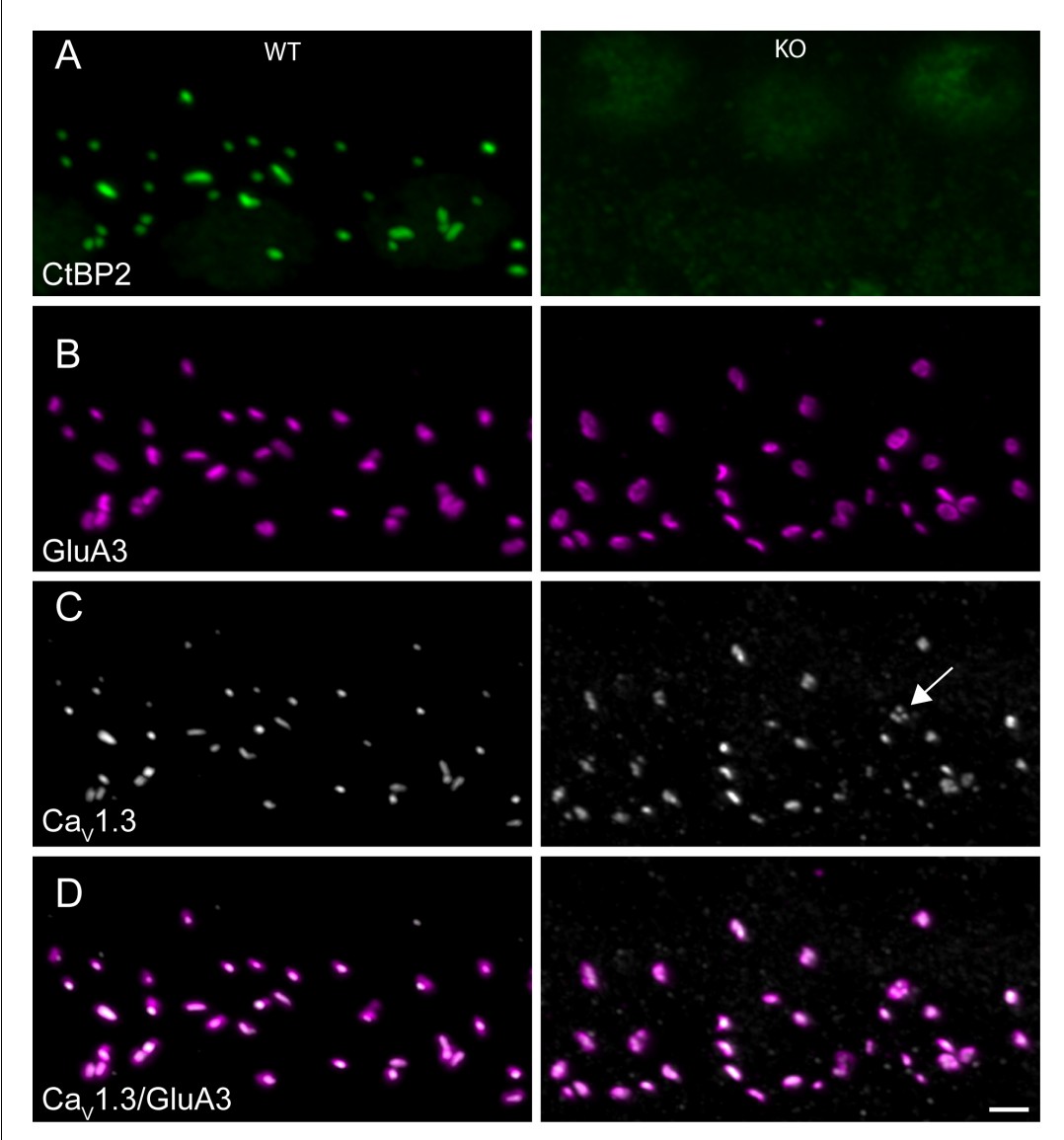

**Figure 3.** Ribbonless synapses contain clusters of presynaptic voltage-gated Ca²⁺channels juxtaposed to postsynaptic glutamate receptors. (A–D) Maximum intensity projections of airyscan z-sections of IHCs in apical cochlea turns of P35 KO and WT littermates, immunolabeled for CtBP2/RIBEYE (A), GluA3 (B), Ca$_V$1.3 (C) and overlay between calcium channels and glutamate receptors (D). IHC afferent synapses from WT have a synaptic ribbon: a cytoplasmic presynaptic density comprised mainly of RIBEYE protein, recognized by anti-CtBP2 (green). The closely associated presynaptic voltage-gated Ca²⁺ channels (Ca$_V$1.3, gray) and postsynaptic AMPA receptors (GluA3, magenta) are co-labelled. IHC afferent synapses from KO lacked synaptic ribbons. The AMPA receptors in KO mice exhibited a ring-like morphology, and the Ca$_V$1.3 channels clustered in the form of round spots or elongated stripes, both similar to morphological features of WT synapses. Scale bar: 2 μm.

DOI: https://doi.org/10.7554/eLife.30241.005

n = 13, p=0.51) or GluA3 (15 ± 3, n = 9 vs. 15 ± 2, n = 13, p=0.43) (**Figure 4A,B**). There was also no difference in the number of paired synaptic puncta between genotypes, with values of 14 ± 2 (n = 9) for WT and 13 ± 2 (n = 13) for KO (p=0.13). Thus, the presence of the ribbon did not influence the number of apparently functional synapses in each IHC.

Careful inspection of GluA3 immunoreactivity showed that the glutamate receptor puncta were larger on average in the KO (**Figure 4D**). Puncta volumes were measured for presynaptic markers Ca$_V$1.3 and Bassoon as well as for postsynaptic markers Homer and GluA3 (**Figure 4C**). Like GluA3 puncta, Homer puncta were significantly larger on average in the KO (KS-test, p≤0.0001) (**Figure 4D**). The volumes of the Ca$_V$1.3 and Bassoon distributions were very similar among

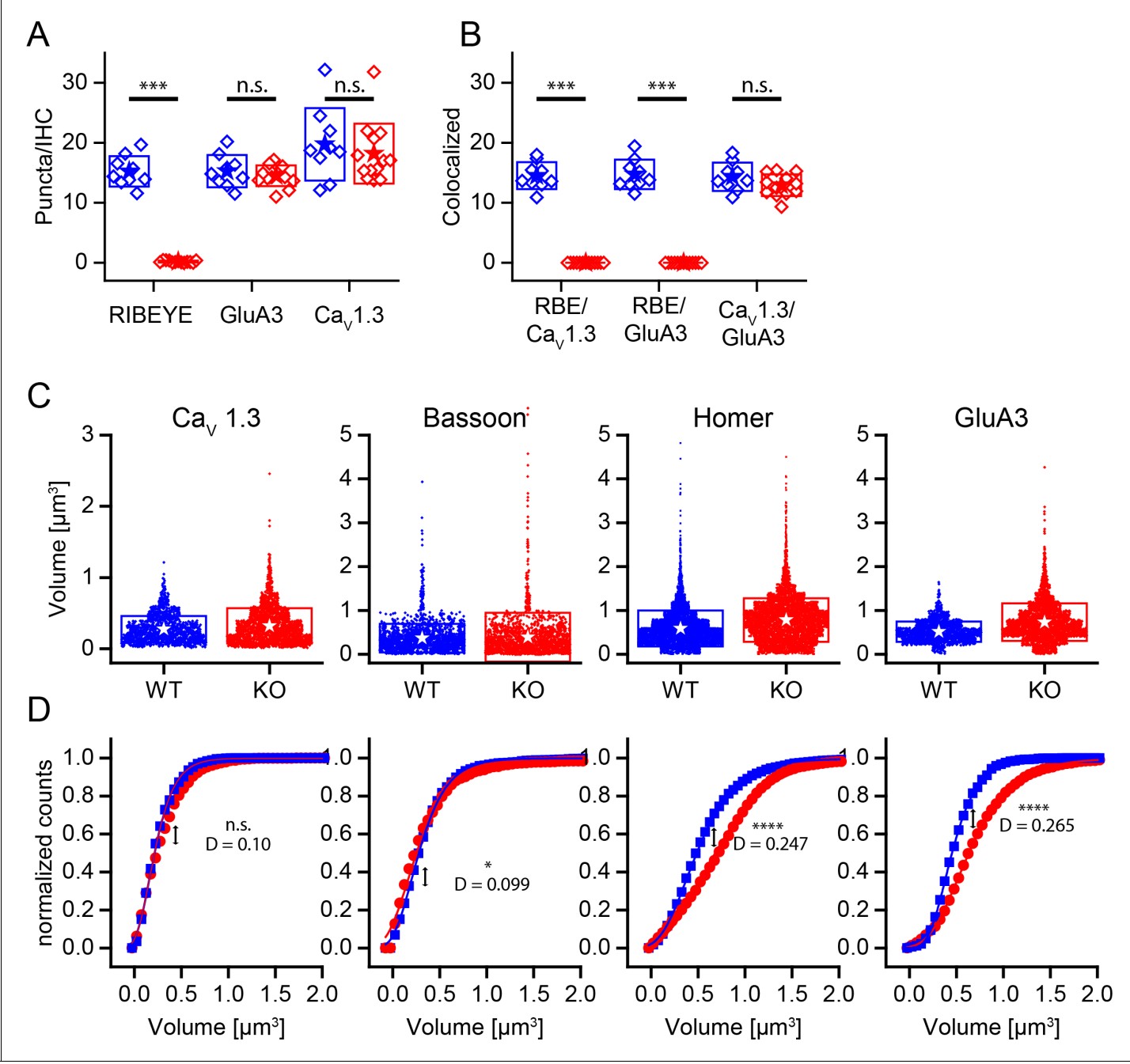

**Figure 4.** Changed size distribution but unaffected synapse number and organization in ribbonless synapses. (**A**) Quantification of RIBEYE, GluA3 and Ca$_V$1.3 puncta in the organ of corti normalized to the number of IHCs in analyzed regions of interest. The number of GluA3 receptor and Ca$_V$1.3 puncta is unaffected (p>0.05). WT in blue, n = 9 images from three mice, 95 cells, quantified puncta — RIBEYE = 1456, GluA3 = 1459, Ca$_V$1.3 = 1897. KO in red, n = 13 images from three mice, 248 cells, quantified puncta — RIBEYE = 28, GluA3 = 2149, Ca$_V$1.3 = 2681. (**B**) Colocalization of synaptic puncta per IHC. In the absence of RIBEYE, the number of synapses, defined by colocalization of postsynaptic GluA3 and presynaptic Ca$_V$1.3, is unaffected (p=0.13) in KO. (**C**) Volume of presynaptic Ca$_V$1.3, Bassoon and postsynaptic Homer and GluA3 in WT (blue, n synapses — Ca$_V$1.3 = 1370; Bassoon = 1917; Homer = 5219; GluA3 = 1611) and KO (red, n synapses — Ca$_V$1.3 = 2132; Bassoon = 1817; Homer = 4990; GluA3 = 2334) mice. (**D**) Normalized cumulative distributions of volumes show increased sizes of postsynaptic GluA3 and Homer immunoreactivity. (**C**) Boxes represent standard deviations of the mean. Significance levels of two-tailed unpaired t-tests (**A, B**) or two sample Kolmogorov-Smirnov test (**C**): (**D**) represents the maximum distance of the Kolmogorov-Smirnov distribution functions. n.s., not significant; *p≤0.05; **p≤0.01; ***p≤0.001; ****p≤0.0001.

DOI: https://doi.org/10.7554/eLife.30241.006

genotypes, but the volume of Bassoon was significantly higher in KO. Most WT and KO synapses exhibited the typical elongated stripe-like appearance of $Ca_V1.3$ puncta. However, we noted a tendency for KO synapses to have $Ca_V1.3$ clusters that were fragmented into smaller puncta (*Figure 3C*, arrow; see also Companion Paper). These anatomical data support the hypothesis that the ribbon is not a major regulator of synapse formation or stability; however, it does perhaps suggest that postsynaptic receptors are sensitive to activity, and the increased plaque size might reflect a compensation for reduced activity.

## Basolateral IHC properties

We compared the properties of the IHC voltage-gated $Ca^{2+}$ current (*Figure 5*), which drives exocytosis. We first blocked outward $K^+$ currents using $Cs^+$ and tetraethylammonium (TEA) in the internal solution to isolate $Ca^{2+}$ currents in WT and KO IHCs (*Figure 5A,B*). Current-voltage plots showed no difference in the voltage of half-activation ($V_{1/2}$) or maximum current amplitude (*Figure 5C–E*). Half activation voltages were $-34 \pm 0.3$ mV (n = 18, WT) and $-33.4 \pm 0.3$ mV (n = 26, KO) for animals P10–P13. The companion paper found small changes in the voltage-dependent properties of the calcium channels at older ages and in higher external calcium, differences that might suggest compensatory effects of ribbon loss or later specializations that are ribbon dependent. Examples of current elicited with $K^+$ as the major internal cation and without TEA are presented in *Figure 5F,G*. Recordings were made at P20–P24. No differences were observed in maximal current or in $V_{1/2}$ (*Figure 5H*). Zero-current potentials monitored in current-clamp mode (*Figure 5I*) were not different, with means of $-63 \pm 10$ mV (n = 5, WT) and $-65 \pm 8$ mV (n = 8, KO).

## Release is slower and smaller for naturalistic stimuli in the absence of the ribbon

We measured presynaptic exocytosis with the two-sine wave method for tracking vesicular fusion by monitoring real-time changes in cellular membrane capacitance (Cm) from IHCs (*Santos-Sacchi, 2004*; *Schnee et al., 2011a*). The two-sine method allows continuous monitoring of Cm during the depolarizing stimulus as compared to a before and after monitoring of capacitance changes, which is particularly helpful during smaller, longer depolarizations (*Schnee et al., 2011b*). Typically, small depolarizations lead to relatively slow, linear increases in Cm during depolarization. In contrast, larger depolarizations induce an abrupt departure from linearity termed the 'superlinear' response (*Schnee et al., 2011b*). Both types of responses were present in recordings from the IHCs of WT and KO mice alike (*Figure 6A,B*).

From Cm measurements, we estimated the sizes of synaptic vesicle pools, their rates of release and the $Ca^{2+}$-dependence of release. In response to strong depolarizations, we found no difference in maximal release between genotypes for either the linear or superlinear release components (*Figure 6C*). The linear component was $52 \pm 24$ fF (n = 15, WT) compared to $42 \pm 22$ fF (n = 15, KO); and the superlinear component of release was $223 \pm 79$ fF (n = 17, WT) compared to $253 \pm 116$ fF (n = 17, KO). Likewise, no differences in release rates (*Figure 6D*) were observed for the linear component ($101 \pm 72$ fF/s [n = 15, WT] compared to $75 \pm 50$ fF/s [n = 14, KO]) or for the superlinear component ($256 \pm 164$ fF/s [n = 17, WT]) and $253 \pm 117$ fF/s [n = 18, KO]). In addition, the $Ca^{2+}$-dependence of release for the linear component was not different ($0.9 \pm 0.03$ fF/pC, n = 37, for the WT as compared to $0.93 \pm 0.03$ fF/pC, n = 34, for the KO). The voltage at which the superlinear component of release first appears was also not different between the two mouse groups (*Figure 6E*). Fitting the data with a simple Gaussian function gave values of $-40 \pm 9$ mV for WT and $-41 \pm 3$ mV for KO for the center values. However, the widths of the functions were different, with values of $7.6 \pm 2$ for WT compared to $14.3 \pm 3$ for KO (p<0.002). Thus, although the variance in release properties increased for the KO animals, there were no major differences in the ability of the synapses to respond to strong stimuli.

In contrast to the comparable maximal responses, smaller depolarizations revealed a reduction in release at early time points in the KO. The difference observed was masked at longer time points or by stronger depolarization (*Figure 6F–J*). Comparing capacitance changes at short intermediate and longer time points in response to smaller depolarizations showed that the KO consistently had reduced release as compared to the WT. Fitting the data in *Figure 6G,H* with a simple exponential demonstrated that, although both genotyes reached the same values of release with strong

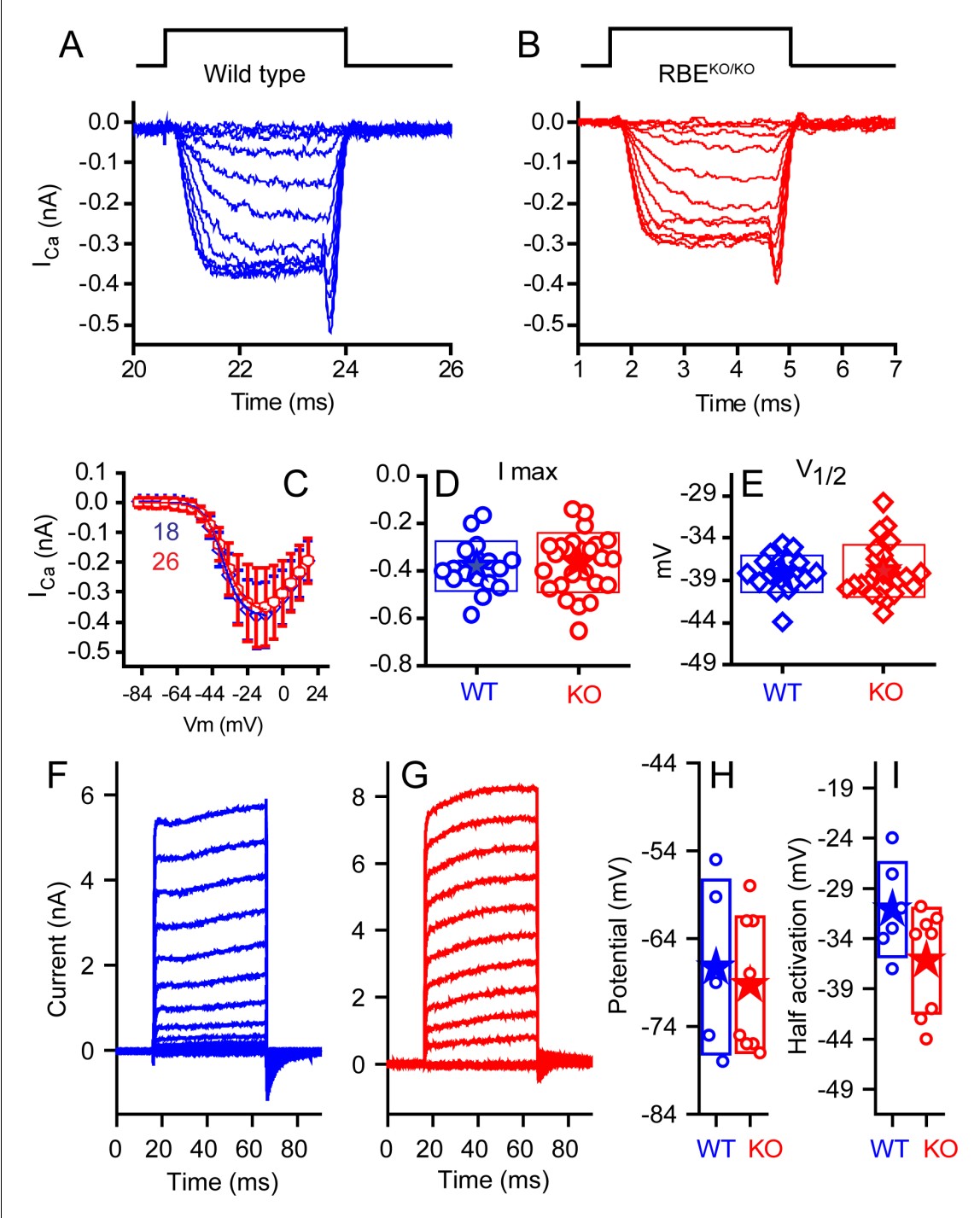

**Figure 5.** No differences were observed in calcium or potassium currents between KO and WT animals. $Ca^{2+}$ currents obtained from mice ranging in age from P10–P12 for WT (A, blue) and P10–P13 KO (B, red), with time course of stimulus shown above, are comparable in all measured properties. Voltage-current plots, presented in (C) as mean ± SD show no difference (n provided in figure). Summaries of peak current (D) and voltage of half activation ($V_{1/2}$, E) are not statistically different (n = 18 for WT and n = 26 for KO, p=0.13 for peak current and 0.326 for $V_{1/2}$). Potassium currents, obtained from responses to 50 ms voltage pulses ranging from −124 mV to 16 mV in 10 mV increments, from a holding potential of −84 mV are presented in (F) for WT and (G) for KO mice at P24. A summary of the half activation voltage as estimated from a Boltzmann fit to the tail current voltage-current (H) found no statistical difference (n = 6 and 8 for WT and KO, respectively, with p=0.084). Current-clamp measurements found no difference in resting potential (I) (n = 5 and 8 for WT and KO, respectively, p=0.69).

DOI: https://doi.org/10.7554/eLife.30241.007

depolarizations, the relationships varied in two ways. For the shortest time point, release started at −40 mV as compared to −45 mV, with all other variables remaining constant. Given that the numbers for release are quite small and variable, conclusions should be conservative, but support the argument that fewer vesicles are available for release at this early time point and that recruitment is minimal. For the longer time point *Figure 6H*, the voltage-dependence (Cm/Vm) varied between groups, being $7.3 \pm 3.6$ for WT as compared to $12.7 \pm 5.8$ for KO. F-tests show these fits to be different at the p<0.002 level. Longer time points, as depicted in *Figure 6I,J*, show no difference between groups. In addition, for smaller depolarizations, the delay to release onset was greater in the KO IHCs, consistent with a reduction in the number of vesicles immediately available for release (*Figure 6K*). For example, depolarizations to −44 mV resulted in significantly shorter latency in WT ($102 \pm 98$ ms, n = 16) compared to KO ($204 \pm 166$ ms, n = 17, p=0.039). Stronger depolarizations (>−39 mV) resulted in no difference in release latency. The time constants of endocytosis were not significantly different ($6.0 \pm 4$ s, n = 13, WT and $7 \pm 5$ s, n = 13, KO). Together these data suggest that fewer vesicles were available for release at ribbonless synapses, but that with stronger depolarizations or longer time courses of depolarization, the recruitment of additional vesicles masked this immediate difference in release latency and amplitude.

The ribbon may be a barrier to $Ca^{2+}$ diffusion, creating a restricted space for sharp $Ca^{2+}$ nanodomains beneath the ribbon (*Roberts, 1994*; *Graydon et al., 2011*). Intracellular $Ca^{2+}$ buffering can alter the release properties, depending on the relative distributions of $Ca^{2+}$ channels and release sites. In the retina, the KO increases buffering efficacy (*Maxeiner et al., 2016*), suggesting looser coupling between $Ca^{2+}$ channels and release sites. To investigate the effects of ribbon absence on stimulus-secretion coupling in P11–P13 IHCs, we compared between genotypes the effects on release properties of changing concentration of the slow $Ca^{2+}$ buffer EGTA (*Figure 7*). Examples of release obtained with either 0.1 or 5 mM EGTA are presented in *Figure 7A* (control experiments all use 1 mM EGTA). Although increasing buffer concentration reduced release and lowering buffer increased release, the relative changes were comparable between WT and KO mice for release measured at 1 s (*Figure 7B*). Data previously demonstrated that increasing $Ca^{2+}$ buffering increased the time to superlinear release (*Schnee et al., 2011b*). *Figure 7C* demonstrates that this effect is retained in the KO and that there are no major differences between WT and KO modulation by $Ca^{2+}$ buffering. These results contrast with findings from retinal bipolar cells and in the zebrafish KO (*Maxeiner et al., 2016*; see Discussion). Similarly, release rates measured for steps to −45 mV were not significantly different for the initial release components (*Figure 7D*). We further investigated the first component of release, typically thought to represent vesicles in early release pools, not requiring trafficking. Comparing release at two voltages again showed that the buffer greatly altered release properties but not in a way that differed between genotypes. However, it should be noted that the difference in early release observed in *Figure 6* with 1 mM EGTA was lost with 0.1 mM EGTA. Thus, it is possible that a sensitivity exists that is below a clear detection limit for our capacitance measurements.

## EPSCs are more heterogeneous in size and kinetics in the absence of the ribbon

$C_m$ measurements are a powerful tool for investigating whole-cell exocytosis but they lack sensitivity (1 fF reflects fusion of ∼20 vesicles), they reflect information from all synapses and they can be quite variable. To obtain a more refined view of activity from individual synapses, we recorded postsynaptically, from the afferent fiber bouton. Hair cells bathed in artificial perilymph lack the standing inward current through mechanoelectric transduction channels in hair bundles, causing the resting membrane potential to be relatively hyperpolarized (*Farris et al., 2006*; *Johnson et al., 2012*). To depolarize hair cells and increase the rate of vesicle fusion, we applied a high potassium (40 mM $K^+$) solution to depolarize the hair cell. $K^+$ application to the hair cell in current-clamp mode depolarized hair cells by an average of $32 \pm 5$ mV (n = 8; *Figure 8A*). There was no difference in depolarization between genotype, thus allowing conclusions regarding postsynaptic responses to be ascribed to presynaptic effects.

Afferent boutons of type I auditory nerve fibers were recorded ex vivo at ages of P17–P21. Owing to the approach vector of the recording pipette, recordings were made from boutons located on the side of the IHC facing the ganglion (modiolar face) rather than the pillar cell side. The start of the high $K^+$ response was defined as the point at which the mean EPSC rate exceeded the rate in

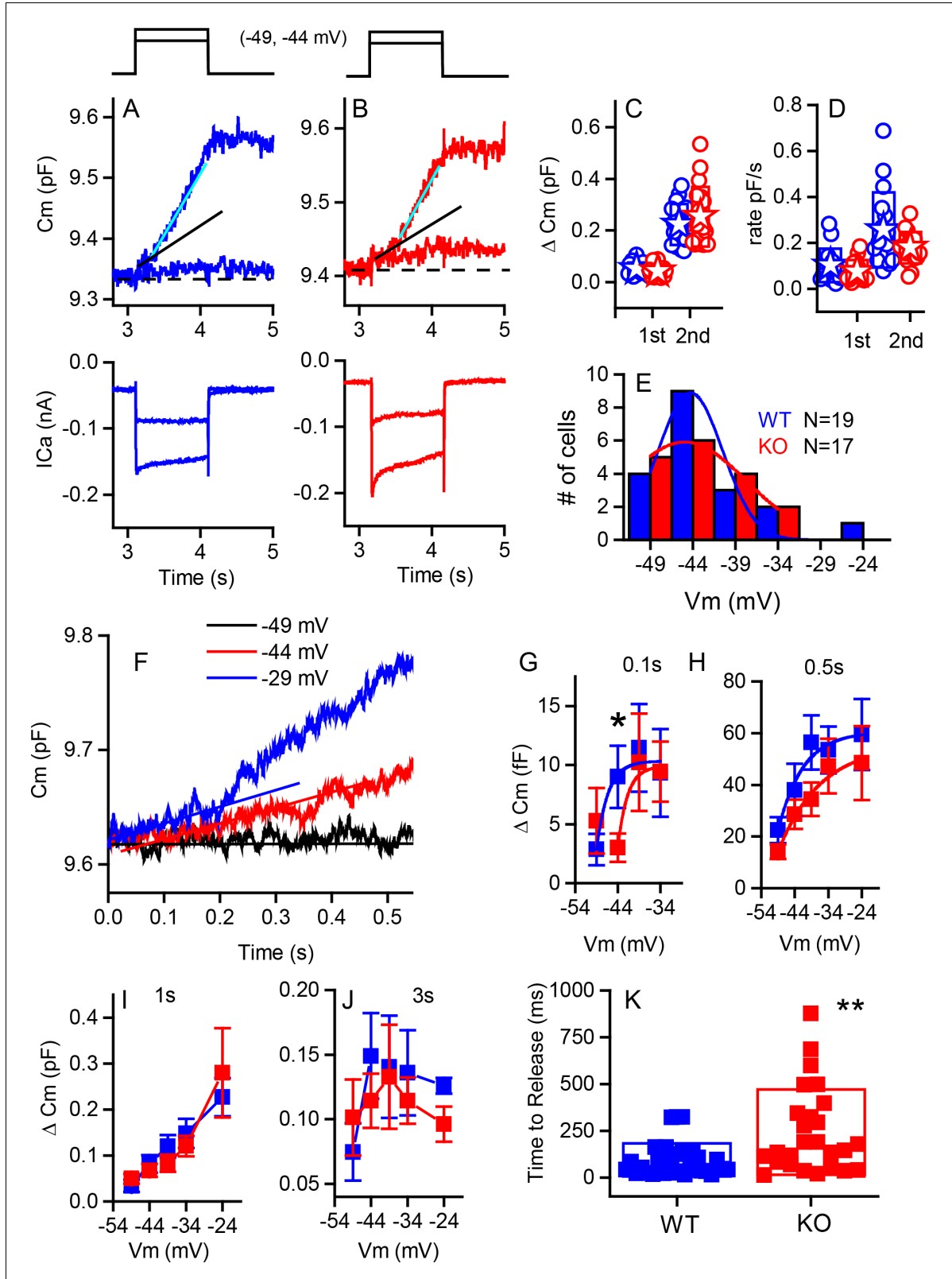

**Figure 6.** Capacitance responses recorded from RIBEYE knockout mice show little difference between phenotypes. Example records of Cm responses (top), showing two release components and $Ca^{2+}$ currents (below) recorded from a WT (blue) (**A**) and a KO (red) mouse (**B**) in response to 1 s depolarizations from −84 mV to −49 and −44 mV. The dashed line indicates pre-stimulus baseline Cm. The black solid line indicates the first linear component of release, and the cyan line the second superlinear component. (**C**) Summary plot of first and second component magnitudes. No

*Figure 6 continued on next page*

*Figure 6 continued*

statistical difference was found between groups. For the linear component, n = 15 for WT and 15 for KO, p=0.23. For the second superlinear component, n = 17 for both groups and p=0.38. Filled stars indicate mean for this and all subsequent figures. (D) Summary plot of first and second component rates obtained by linear fits to data points, as indicated by lines in (A,B). No statistical difference was found between groups where, n = 16 for WT and 13 for KO, with p=0.266. For the second component rate, n = 17 for both groups, with p=0.085. (E) A population histogram for the voltage at which the superlinear response is first observed. Fits to Gaussian distributions show no change in the mean value but an increase in the width of the plot for the KO. (F) Example of delay in onset of capacitance change with small depolarizations for a KO cell. Plots of Cm verses voltage responses measured at 0.1 (G), 0.5 s (H) 1 s (I) or 3 s (J) show a significant difference for release at 0.1 s and −44 mV (p=0.03), error bars represent SEM. Fitting the relationships with simple exponential functions reveals differences in the shapes of the curves but not in maximum release for the 0.1 and 0.5 plots, whereas there are no differences in curve shapes for the 1 and 3 s time points. N provided in figure for each group. (K) Shows that time to release is significantly longer for KO compared to WT for depolarizations eliciting less than 50% of the maximal calcium current; n = 32 for WT and 26 for KO, p=0.0012.

DOI: https://doi.org/10.7554/eLife.30241.008

normal external solution by two standard deviations. Examples of postsynaptic responses to K$^+$ applications are provided in *Figure 8B–D*. Two types of responses were observed, sustained responses (*Figure 8B,C*) and transient responses (*Figure 8D*). WT responses were mainly of the sustained type (7 of 8), whereas the KO responses included a smaller proportion of sustained responses (7 of 11). EPSC rates are plotted for transient (*Figure 8E*) and sustained responses (*Figure 8F*) for both WT and KO. EPSC rates in 40 mM K$^+$ perilymph were quantified over 1 s windows at 100 ms intervals and the means are displayed as thicker lines (*Figure 8F*). The peak release rates varied considerably across fiber populations (0–120 s$^{-1}$) and there was a tendency for the KO neurons to reach a lower peak EPSC rate on average: 57 ± 30 s$^{-1}$ (n = 11) for WT compared with 44 ± 31 s$^{-1}$ (n = 18) for KO (*Figure 8G*).

The time to peak EPSC rate was less variable than the absolute rates (*Figure 8H*). The KO had one outlier that steadily increased throughout the K$^+$ application, removal of this point from the statistical analysis resulted in the KO reaching peak EPSC rate sooner (9 ± 10 s, n = 10) than WT (21 ± 12 s, n = 9; p=0.037). Observing the sustained release group shows that both WT and KO reduced EPSC rate over time but that the KO reduced more completely than the WT (*Figure 8G*). 8/11 KO reduced to less than 5% of peak as compared to 3/8 WT. Integrating the rates over time (*Figure 8I*) confirmed an increase in total EPSCs in WT (1.8 ± 1.1 × 10$^3$) compared to KO (0.81 ± 0.66 × 10$^3$, p=0.021).

EPSC amplitude distributions were generated for each recorded fiber. Frequency histograms of EPSC amplitudes could be fit with either a double Gaussian function or a Gaussian and gamma function, where the smallest peak was always fit by a Gaussian function (e.g. *Figure 8J,K*). The area under the smaller peak was calculated and plotted as a percentage of the total in *Figure 8L*, where the KO showed a statistically larger (p=0.034) percentage of small events at 5.5 ± 5.6% (n = 12) as compared to 1 ± 0.7% (n = 7) for WT. The increase in number of small events might suggest less coordination in release because there are fewer vesicles filling release sites. It might also suggest that smaller responses are visible because of an increase in receptor plaque size. Only fibers with more than 100 EPSCs were included for this analysis. No difference was found in the proportion of simple events for each unit, where a simple event is described as having a single rise and decay phase (*Figure 8M*). We also compared the median amplitudes (*Figure 8N*). Although there was a tendency for EPSCs in the KO to be greater in amplitude (212 ± 46 pA, n = 13) than those in the WT (173 ± 36 pA, n = 7), the values were not statistically different. A difference here would support the argument for a postsynaptic compensatory increase in glutamate receptor plaque size. The coefficients of variation (CV) for the amplitude distributions were different on average (p=0.017; *Figure 8O*). The control values are similar to those obtained by *Li et al. (2009)* and *Grant et al. (2010)*. Recordings with a larger proportion of smaller events had larger CV. When small events were removed from the histogram, the CVs were not statistically different between KO and WT. The postsynaptic responses in KO also had a greater proportion of EPSCs > 350 pA (*Figures 8J,K* and *9D*): 1.3 ± 1.3% in WT compared to 8.0 ± 10% for KO EPSCs (t-test for unequal variances, p=0.04). The change in amplitude was not accompanied by a change in kinetics. The increased amplitude might reflect the increase in postsynaptic glutamate receptor plaque (*Figures 3* and *4*).

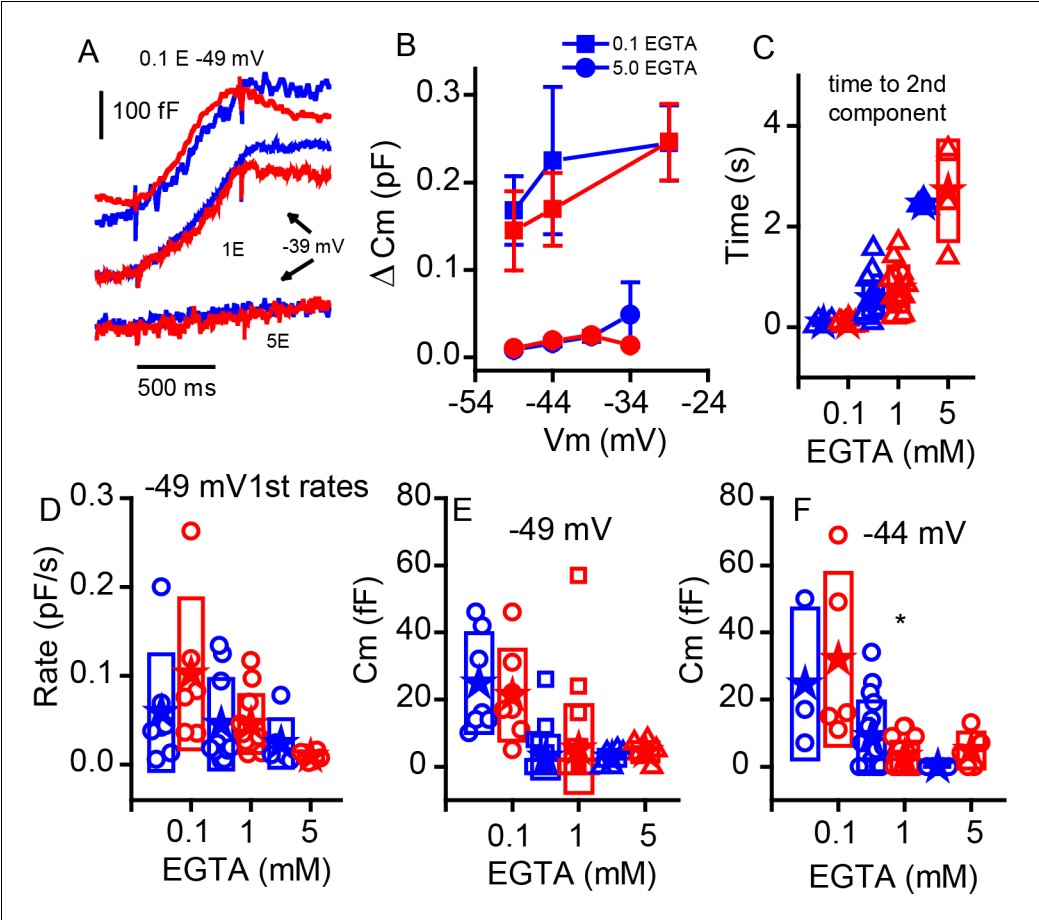

**Figure 7.** Changing internal $Ca^{2+}$ buffering does not distinguish phenotypes. (**A**) Example records of $C_m$ changes in response to 1 s depolarizations from −49 mV in 0.1 EGTA and −39 mV in 1 and 5 mM EGTA for WT (blue) and KO (red). (**B**) Plot of the change in Cm in response to 1 s voltage steps of −49, −44 and −29 mV for 0.1 mM EGTA and −49, −44, −39 and −34 mV for 5 mM EGTA, n provided in figure. (**C**) The time to the second component increases significantly with elevated $Ca^{2+}$ buffer (p-value indicated in figure) but is similar between genotypes: for 0.1 mM EGTA, n = 7 for WT and n = 6 for KO, p=0.93, steps to −29 mV for WT and KO; for 1 mM EGTA, n = 15 for WT and n = 13 for KO, with p=−0.13, steps to −36 ± 8 mV WT and −37 ± 9 mV KO; and for 5 mM EGTA, n = 3 for WT and n = 4 for KO, with p=0.67, steps to −39 ± 5 mV WT and −35 ± 2 mV KO. (**D**) The rate of release to the −49 mV step declines equally in each genotype with increasing EGTA concentration: for 0.1 mM EGTA, n = 7 for WT and n = 6 for KO, p=0.323; for 1 mM EGTA n = 20 for WT and n = 22 for KO, p=0.99; and for 5 mM EGTA, n = 7 for WT and n = 6 for KO, p=0.235. (**E,F**) Box plots of the first component of release for depolarizations to −49 mV (**E**) or −44 mV (**F**). The effects of buffers were similar between the two groups.

DOI: https://doi.org/10.7554/eLife.30241.009

Examples of EPSCs are presented in *Figure 9A* to show the variation in amplitudes and to identify a smaller population of EPSCs with slower kinetics (see inset). As the ribbon is proposed to be involved with multivesicular release (MVR), and the kinetic similarities between large and small EPSCs are a signature of MVR, we investigated the decay of EPSCs in auditory nerve fiber recordings from WT and KO animals. Fitting the EPSC decay with single exponentials provided time constants (*Figure 9B*). Calculating the CVs of EPSC decay time constants (*Figure 9C*) revealed more variability in KO but no significant difference compared to WT. The skewness (*Figure 9F*) of the decay time constant distributions was significantly larger for the KO (0.46 ± 0.2, n = 13) compared to WT (0.29 ± 0.07, n = 7; p=0.04); the KO mean EPSC decay time was greater than the median decay time. In the WT population, 0.44% of EPSCs were <100 pA in amplitude with decay time constants >1 ms, whereas 1.9% of KO events fell into that category. Plotting the decay time constant against amplitude for each EPSC in all fibers (*Figure 9D*) (WT — n = 7 recordings, 15,564 EPSCs; KO — n = 13 recordings, 17,210 EPSCs) identifies a population of small slow EPSCs in the KO that are present in greater proportion in these fibres than in WT fibres (upper left quadrant). The mean

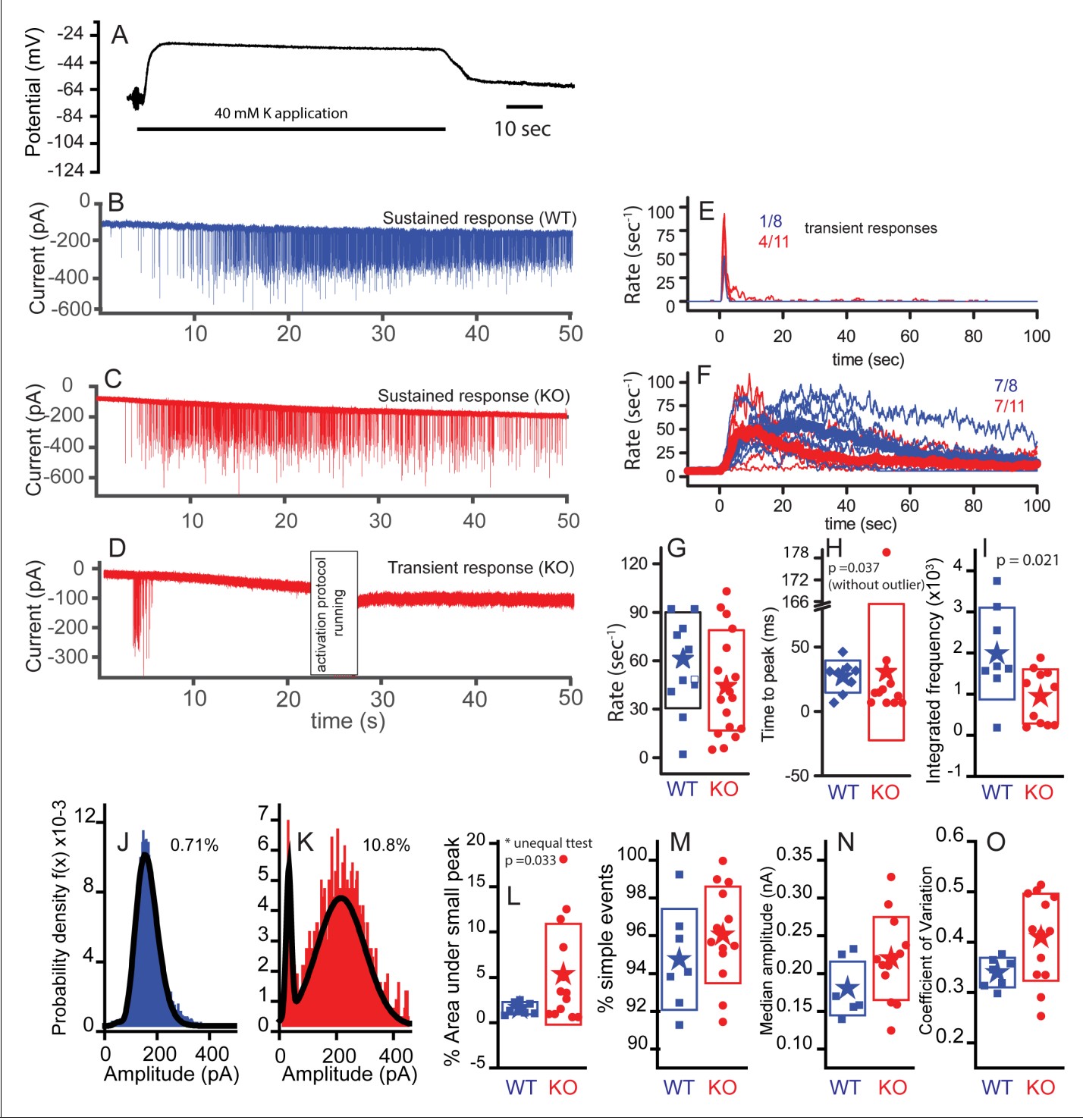

**Figure 8.** Subtle differences exist between WT and KO mice postsynaptic properties. Postsynaptic bouton recordings were obtained from WT or KO in control and during application of 40 mM K$^+$ to the bath. Hair cell recordings demonstrate similar depolarizations in WT and KO, suggesting that any differences in response properties may be synapticaly mediated. (**A**) Cells rapidly recovered following washout of K$^+$. The solid line below the trace indicates the duration of the application. Postsynaptic recordings from afferent boutons in response to 40 mM K$^+$ external solution application are presented in (**B**) for WT and (**C**) for KO with a sustained response and in (**D**) for KO with a transient response. Recordings were made in the presence of 5 μM tetrodotoxin (TTX). Frequency responses were generated using a 1 s window sliding at 100 ms intervals (**E,F**). Populations were defined as transient or sustained by according to whether firing persisted after 20 s. The number of each response type is presented as a fraction in each panel. Maximum frequency responses are presented in box plot format with mean values shown as stars (**G**); n = 10 for WT and n = 18 for KO, p=0.313. The
*Figure 8 continued on next page*

*Figure 8 continued*

time to peak is presented in (H), where onset of $K^+$ application response was defined as the time point when the frequency exceeded the spontaneous frequency +3 x SD for two overlapping time windows (200 ms). The time to peak was statistically different when the one outlier was removed (time point at 181 s); n = 8 for WT and n = 10 for KO. Time to peak only includes fibers whose peak frequency reached 30 Hz, because those with lower peak frequencies had too few points to make an accurate measure. (I) Integrating the frequency response over 50 s shows that WT release in a more sustainable manner than the KO, n = 8 for WT and n = 10 for KO, p=0.021. Representative probability density functions for EPSC amplitudes are presented for WT (number of events = 6583) in (J) and for KO (number of events = 1551) in (K). Solid lines represent fits to either double Gaussian functions or Gaussian and gamma functions (smaller peak always fit with Gaussian function). The curves fitting the small events were integrated to measure the % of events under the small peak (shown next to each example). Only nerves that had at least 100 events are included to ensure the robustness of the fitting. (L) A box plot of the WT and KO percent of area contributed by the small peak. Comparisons using a t-test for unequal variance (Welch test) suggest a significant difference, p=0.032, n = 7 for WT and n = 13 for KO. (M) A box plot comparing the percentages of simple events, defined as single rise and decay times that are not significantly different, n = 7 for WT and n = 13 for KO, p=0.3. (N) Box plots of the median amplitude, where no significant difference was observed, n = 7 for WT and n = 13 for KO for each, p=0.105. (O) Box plots of the coefficient of variation for the distributions in (N). Plots were different, p=0.017, when equal variance was not assumed.

DOI: https://doi.org/10.7554/eLife.30241.010

EPSC decay time constants, when plotted as a function of EPSC amplitude for each recording (bin of 10 pA), illustrate that all of the KO fibers had a proportion of EPSCs with slow time courses, whereas the WT have only two fibers with these slower events (*Figure 9E*). An example of the smaller, slower EPSC is provided in the inset of *Figure 9A*. The distributions of EPSC durations for all EPSCs <100 pA (*Figure 9G,H*) demonstrates the shift toward slower longer duration EPSCs in KO fibers (p>0.001, using the Kolmogorov-Smirnov test). The cumulative probability functions of EPSC duration for all fibers by genotype (thick red and blue lines) illustrate the larger proportion of slow EPSCs in the KO fibers (*Figure 9H*). These data may also support the argument for a larger postsynaptic glutamate receptor plaque, where diffusion of glutamate, coupled with the lower glutamate concentration, might serve to slow the decay kinetics.

## Discussion

The KO mouse provides an unprecedented opportunity to investigate the functional relevance of synaptic ribbons in driving information transfer at the primary auditory synapse. Synapse formation and stability appeared normal when assessed on the basis of synapse counts and protein localization, despite complete absence of the ribbon (*Figures 1*, *3* and *4*). No evidence was found for partial or defective ribbons structures, nor did we detect compensation by other proteins in the absence of RIBEYE (*Figure 1*). Also, cochlear mechanics appeared normal as assessed by DPOAE (*Figure 2*). Thus, this mouse provides an excellent model to investigate the ribbon's role in synaptic transmission.

### Whole-animal function

ABR responses reported a 10 dB threshold shift and about a 50% reduction in ABR amplitude, with no significant effect on the shape of the waveforms or on the delay (*Figure 2*). These data agree qualitatively with the companion paper data, the lack of significance probably being a function of the sample size. The minor auditory deficit may not be expected to produce a change in behavioral sound detection threshold as the ensuing ABR peaks, representing brainstem nuclei, were unaffected. Alternatively, central compensation that masks peripheral deficits might occur. These data suggest that the ribbon is not essential for setting hearing threshold, and that its role in auditory function is more nuanced than, for example, that of otoferlin, which when absent or altered leads to deafness with threshold elevations of more than 100 dB (*Roux et al., 2006*; *Pangrsic et al., 2010*). Further work is needed to better understand ribbon function at the systems level. Additional testing to better investigate auditory processing, such as sound localization experiments, may provide additional insight into how the ribbon might serve systems-level function.

### Synapse formation

Synapse formation is surprisingly normal in KO with an appropriate number of synapses and synaptic markers present (*Figures 1* and *4*). Little change in $Ca^{2+}$-channel distribution or biophysics were

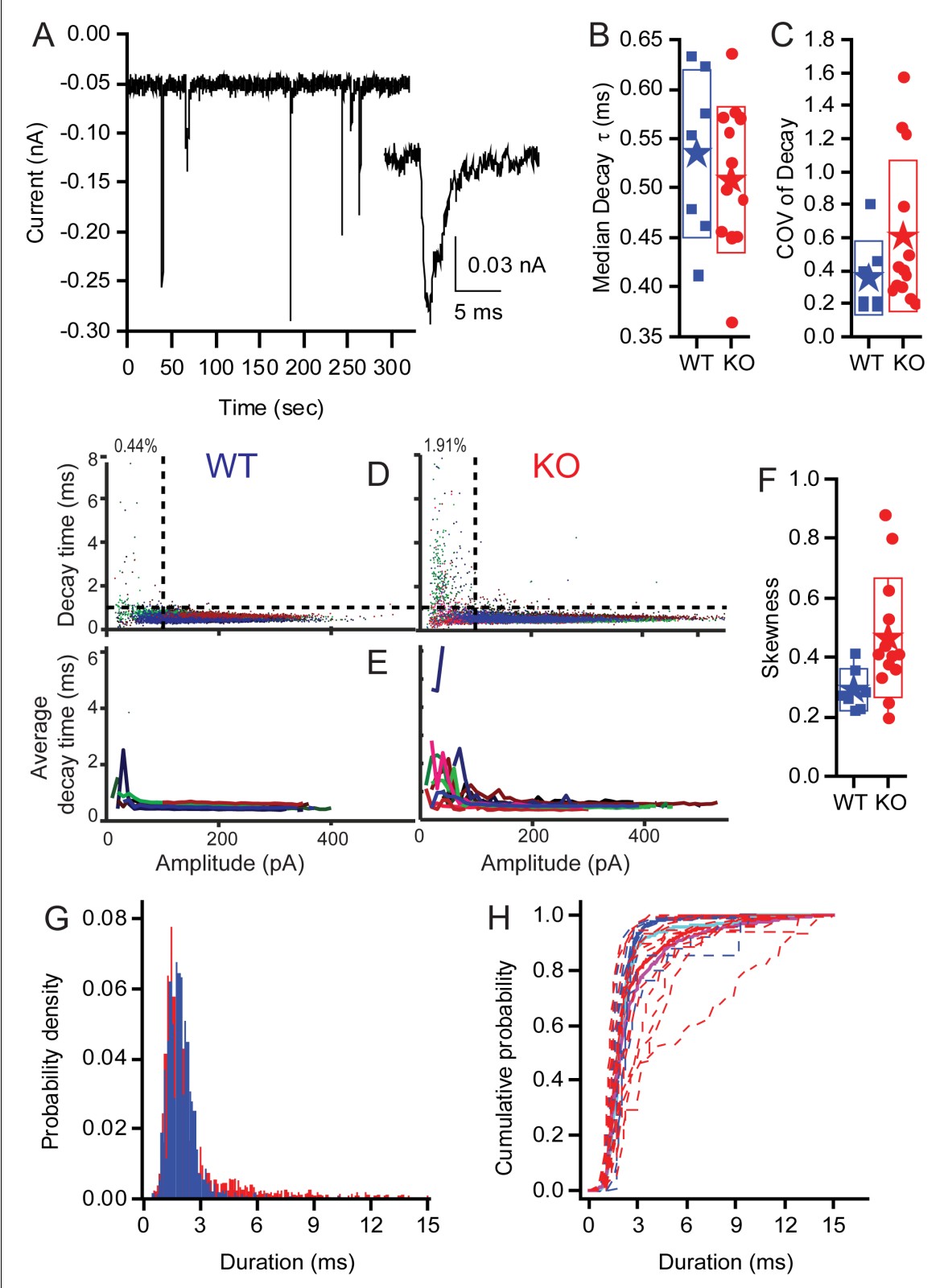

**Figure 9.** KO synapses have a larger population of small slow EPSCs. Seven WT fibres including 15,564 events and 13 KO fibers including 17,210 events were analyzed. (**A**) Examples of EPSCs that range in amplitude and kinetics. The inset highlights one of the small slower EPSCs identified in KO fibers. Median decay times (**B**) and the Coefficients of Variation for the distribution in decay times (**C**) are presented as box plots. Neither population show a statistically significant difference between the WT and KO groups. (**D**) Plots showing the skewness of the distributions of decay times and

*Figure 9 continued on next page*

*Figure 9 continued*
demonstrating that the KO decay time is significantly greater (away from 0), suggesting that the mean is larger than median, largely because they have a long tail of large decay time. (**E**) Plot of the EPSC decay time constant for every fiber (each fiber a different color) for both WT and KO. (**F**) Summary of the data from (C) as average decay time constant with EPSCs binned over 10 pA. (**G**) Pooling fibers for WT and KO recordings allowed for the generation of a probability density function for event duration. These distributions are significantly different as shown by KS test (p<10⁻¹⁸). (**H**) The cumulative probability function for the duration of EPSCs. Dashed lines are the individual fiber distributions. Solid red and blue lines represent the population aggregate (summed responses) for KO and WT, respectively. The cyan and magenta lines are the averages for WT and KO, respectively.
DOI: https://doi.org/10.7554/eLife.30241.011

observed in the KO (*Figure 5*) relative to animals in which Bassoon is disrupted, suggesting that Ca²⁺-channel distribution or regulation does not depend critically on the ribbon (*Frank et al., 2010*). Our companion paper identifies minor changes in Ca²⁺-channel distribution and voltage sensitivity at older ages; but whether these changes are a function of ribbon absence or activity-dependent modulation remains to be explored. The lack of effect on calcium-channel properties and distributions at early ages supports the hypothesis that there is a compensatory mechanism that occurs later during synapse maturation. Given the reports of late development changes in ribbon size and glutamate receptor plaque sizes, activity-driven shaping of the synapses is a reasonable expectation (*Fernandez et al., 2015*). Postsynaptically, both Homer and the glutamate receptors had larger volumes in the KO animals than in WT animals (*Figure 4*). This change also may be an indication of activity-dependent synaptic shaping, where the reduced activity in KO animals results in a postsynaptic upregulation of receptors. The increase in the population of larger amplitude EPSCs in KO animals further supports this conclusion (*Figures 8I,J* and *9D*).

## Comparison to other systems

In zebrafish, a frameshift mutation was used to eliminate RIBEYE from ribbon synapses. This work revealed ghost-like ribbons where vesicles were organized around invisible ribbon-like structures (*Lv et al., 2016*). Neither our investigations, nor the companion paper nor the original work investigating the retina (*Maxeiner et al., 2016*) found comparable ghosts in the KO mouse. We would conclude that either the frame-shift approach did not account for all RIBEYE or there is a strong compensatory mechanism in zebrafish that is not found or is much weaker in mammals. We investigated CTBP1 as a potential compensatory molecule, but as *Figure 1* shows, there is no labeling near synapses in the KO mouse. The presence of ribbons in photoreceptors of zebrafish KO support the idea that there are fundamental differences between zebrafish and mammalian systems. In addition, work in zebrafish suggests a tight reciprocal relationship between calcium-channel numbers, calcium-current amplitude and ribbon size (*Lv et al., 2016*). As no changes in calcium-channel distribution or numbers were observed in the KO animal, despite the loss of the ribbon, this relationship does not appear to exist in mammals or is greatly reduced, again supporting fundamental differences between model organisms.

## Mechanism of ABR amplitude reduction

The decrease in ABR amplitude is probably a function of the reduced frequency of EPSCs, leading to the generation of fewer action potentials (*Figure 8F*). In addition, the increase in proportion of EPSCs that have lower amplitudes and longer durations (*Figure 9H*) is expected to increase the likelihood of spike generation failure (*Rutherford et al., 2012*). As larger EPSCs are better phase-locked than smaller EPSCs (*Li et al., 2014*) and present works finds a 5–10% increase in smaller EPSCs, one might predict a lower vector strength and reduced phase locking. In fact, our companion paper does observe these differences in single-unit firing. Similarly, the increased population of smaller EPSCs is reminiscent of the morphology seen in young animals when compared to older animals and so might reflect a developmental delay in the KO. Fewer fibers fire together at sound onset, reducing auditory nerve synchrony and wave 1 amplitude (*Figure 2C,D*). Measurements of exocytosis from IHCs in response to small depolarizations, clearly showed a reduction in release and an increase in time to release in the KO (*Figure 6E,F*). This suggests a decrease in the number of vesicles available at the synapse for immediate release (which is supported by TEM counts). The reduction in available vesicles for release, as well as the population of reduced amplitude EPSCs, might also predict a spike-timing deficit as described in the companion paper. However, the more rapid response

in WT was masked by stronger stimulation (*Figure 6F*), indicating that vesicle replenishment remained robust despite the absence of the ribbon-associated synaptic vesicle pool. Given the robust nature of vesicle trafficking, the primary role for the ribbon may be in ensuring that a population of vesicles is present for the timing of onset responses, whereas vesicle trafficking can serve to replenish vesicle populations during sustained release. A similar result was reported by *Maxeiner et al. (2016)* in retina.

Postsynaptic recordings suggest that the KO reaches peak release sooner than WT and at a similar peak rate, implying that vesicle numbers for immediate release are comparable. By contrast, TEM suggests that fewer vesicles are available at the synapse, and the small increase in the population of small EPSCs also suggests a reduction in filled release sites. Thus, it is possible that the ribbon acts as a barrier for vesicle interaction with the release site, while simultaneously serving to increase the total number of available vesicles. Interestingly, this idea was first suggested by *Furukawa and Matsuura (1978)* and later supported by *Schnee et al. (2013)* as a possible mechanism for explaining neural adaptation. However, the companion paper suggests modifications at the synapse that might also be in accord with the postsynaptic measurements reported here, that is, more vesicles at the membrane. If there is an increase in presynaptic density area and more vesicles per active zone, it might be predicted that faster immediate release occurs but sustainability might be reduced, EPSC data also supports this idea. To resolve these possibililities, we need a better understanding of the molecular underpinnings of the active zone and the number of available release sites.

## Synaptic vesicle replenishment for sustained release

Our data argue against a role for the ribbon in vesicle formation and replenishment. Total exocytosis, from both the linear and superlinear components, was not reduced in the KO, implying that a similar total number of vesicles could be recruited for release (*Figures 6A,B and* and *7A*). In contrast to the zebrafish lateral line (*Lv et al., 2016*), we measured no change in vesicle size between KO and WT, indirectly supporting our view that the ribbon is not regulating or involved in vesicle formation, a result corroborated by the companion paper. Thus, no change in vesicle size or maximal release supports the conclusion that the ribbon is not involved in vesicle formation.

## $Ca^{2+}$ influx and coupling to exocytosis

The ribbon may act as a barrier to $Ca^{2+}$ diffusion, creating a higher local $Ca^{2+}$ concentration that synchronizes the fusion of multiple vesicles (*Graydon et al., 2011*). In this case, the ribbon's presence would reduce the sensitivity of exocytosis to exogenous $Ca^{2+}$ buffers (by elevating concentrations of $Ca^{2+}$). Likewise, in retina from KO mice, EGTA reduced spontaneous release in the KO but not in WT (*Maxeiner et al., 2016*). In contrast, our data shows that buffering efficacy was not altered in the KO (*Figure 7*), suggesting that the ribbon does not play a major role in the regulation of intracellular $Ca^{2+}$ concentration at IHC active zones. The similar response in WT and KO might indicate a small effect, but at a level below a clear detection limit for capacitance measurements. It is possible that the geometricand synaptic architecture differences between retina and cochlea alter the sensitivity to $Ca^{2+}$ buffers or that the effect noted in retina is indirect, for example EGTA-AM might alter the presynaptic cell's resting potential. Thus, our results are not fundamentally different from those reported by *Maxeiner et al. (2016)*, and both support the idea that the ribbon serves to consolidate vesicles near to release sites as suggested by *von Gersdorff et al. (1996)*.

## Ribbon and multiquantal release

The underlying mechanism for multiquantal release remains unknown but might be driven by simultaneous multiquantal or multivesicular release (*Glowatzki and Fuchs, 2002*). The majority of EPSCs measured in the KO remain large and uniformly fast in onset and decay times and the EPSC amplitude is not reduced, suggesting that the ribbon does not play a pivotal role in multivesicular release. Thus, mechanisms that involve sequential fusion of vesicles along the ribbon, and then release or rapid sequential vesicle release that piggybacks onto a release site from the ribbon, are unlikely to be valid as these should be dramatically altered by the loss of the ribbon. One remaining possibility is the synchronous release of vesicles from multiple release sites. Although the ribbon is not required for this mechanism, it may indirectly modulate this mechanism by influencing the number of docked vesicles available for release. The increase in the number of smaller slower EPSCs, probably arising

from the availability of fewer docked vesicles for synchronous release, may support a ribbon role; however, the small relative contribution of this vesicle pool indicates a minor role. The phenomenon of large EPSC amplitude variability may still be explained by a univesicular mechanism that includes a dynamic fusion pore, which presumably would not require a ribbon (*Chapochnikov et al., 2014*). Where identified, the slow foot-like response described by *Chapochnikov et al., (2014)* did not differ among genotypes.

### Comparison to Bassoon KO

In mice, the effects of KO are subtle compared with those of disruption of Bassoon protein. Loss-of-function mutation in Bassoon led to a reduction in the number of IHC ribbons associated with presynaptic elements, ABR threshold elevation of 30 dB, longer ABR wave 1 latency and first-spike latency, and fewer $Ca^{2+}$ channels at active zones (*Khimich et al., 2005*; *Buran et al., 2010*; *Frank et al., 2010*; *Jing et al., 2013*). Thus, the effects of loss of Bassoon protein (and membrane-anchored ribbons) were greater than those of loss of RIBEYE protein, consistent with a more critical role for Bassoon in scaffolding synaptic proteins. Also, given Bassoon's presence at efferent synaptic endings in the organ of Corti, it is not surprising that the functional loss of Bassoon has a greater phenotype than that of RIBEYE loss (Supplemental *Figure 1*).

### Summary

A central tenet for ribbon function is maintaining a large pool of vesicles near the synapse. The ribbon may serve as a conveyor belt that both maintains vesicles and traffics vesicles to release sites (*Parsons et al., 1994*; *Nouvian et al., 2006*). Our data suggest that the ribbon increases the time that a vesicle remains in the synaptic region. Robust vesicle trafficking and replenishment, via either the recycling or transport of new vesicles to the synaptic zone, provides vesicles to the synaptic zone, but the time that vesicles remain near to release sites may be shortened in the absence of the ribbon that serves to maintain them close to the synapses. A model for this basic idea has been proposed (*Graydon et al., 2014*) that reproduced data from bipolar cell synapses. The addition of a more robust trafficking mechanism may allow this model to also reproduce hair cell synapses. The increase in variance was greater for KO, whether measured as coefficients of variance (CV), skewness or simply standard deviation in measured parameters, such as time to release, release rate and EPSC amplitude. The increased variance may simply reflect a nonuniform population of vesicles near the synapse in the absence of the ribbon. Thus, the ribbon may serve as a biological buffer, maintaining a uniform population of vesicles near release sites.

## Materials and methods

### Animals

Mice of both sexes from a mixed genetic background (C57bl6, 129/Ola) were bred from heterozygous breeding pairs and genotyped by PCR for the knockout of the A-domain of*Ctbp2* (*Maxeiner et al., 2016*), while wild-type littermates served as controls. The Administrative Panel on Laboratory Animal Care at Stanford University (APLAC #14345) and the Animal Care and Use Committee at Washington University in St. Louis approved all animal procedures. Experiments were performed blinded to genotype.

### Auditory measurements

We measured auditory brainstem responses (ABRs) and distortion products of optoacoustic emissions (DPOAEs) in mice of postnatal day 21 as previously described (*Oghalai, 2004*). Mice were anesthetized by administering 100 mg/kg ketamine and 10 mg/kg xylazine intraperitoneally and their body temperature held constantly at 37°C (FHC, DC-temperature controller and heating pad) until they fully recovered.

Sound stimuli were generated digitally by a self-written software (*Oghalai, 2004*), written in MATLAB (version 7.0, MathWorks), that controlled a custom-built acoustic system using a digital-to-analog converter (National Instruments, NI BNC-2090A), a sound amplifier (TDT, SA1), and two high-frequency speakers (TDT, MF1) *Oghalai, 2004*. The speakers were connected to an ear bar and

calibrated in the ear canal prior to each experiment using a probe-tip microphone (microphone type 4182, NEXUS conditioning amplifier, Brüel and Kjær).

ABRs were recorded by placing three needle electrodes subcutaneously at the vertex, below the left ear, and a ground electrode close to the tail. The signals were amplified 10,000 times using a biological amplifier (Warner Instruments, DP-311) digitized at 10 kHz, and digitally band-pass filtered from 300 to 3,000 Hz. The stimulus for eliciting the ABR was a 5 ms sine wave tone pip with $\cos^2$ envelope rise and fall times of 0.5 ms. The repetition time was 50 ms, and 260 trials were averaged.

At each frequency, the peak to peak voltages of ABR signals, at stimulus intensities ranging from 10 to 80 dB sound pressure levels (SPL), were measured in 10 dB steps and fitted and interpolated to find thresholds five standard deviations above the noise floor. For statistical purposes, we defined threshold as 80 dB SPL, if no ABR was detected.

## Distortion products of otoacoustic emission (DPOAE)

For stimulation of DPOAEs, two sine wave tones of equal intensity ($l_1 = l_2$) and 1 s duration were presented to the ear. The tones ranged from 20 to 80 dB SPL attenuated in 10 dB increments and the two frequencies ($f_2 = 1.2f_1$) ranged from $f_2$ with 4–46 kHz. The acoustic signaldetected by the microphone in the ear bar was sampled at 200 kHz and the magnitude of the cubic distortion product ($2f_1$–$f_2$) determined by FFT. Averaging 20 adjacent frequency bins surrounding the distortion product frequency determined the noise floor. DPOAE thresholds were reached when the signal was greater than three standard deviations above the noise floor.

## Tissue preparation, staining and confocal microscopy

For staining of pre- and postsynaptic markers, we used antibodies against Homer (Synaptic Systems, 160003, 1:800 dilution), Bassoon (AbCam, ab82958, 1:500 dilution), CtBP1 (BD-Biosciences, 612042, 1:600 dilution), CtBP2/RIBEYE (Santa Cruz Biotechnology, sc-5966, 1:300 dilution; BD Transduction Lab, 612044, 1:400), $Ca_V1.3$ (Alomone Labs, ACC-005, 1:75), and GluA3 (Santa Cruz Biotechnology, sc-7612, 1:200 dilution). Cochlea of P22 or P35 mice were isolated in HBSS solution or artificial perilymph, a small hole was introduced at the apex, cochlea were gently perfused through round and oval windows with freshly prepared 4% (w/v) PFA (Electron Microscopy Sciences) in PBS until the fixative became visible at the apex. Organs were incubated in the same fixative solution for 30 min at room temperature. Whole-mount organs of Corti were microdissected and the tectorial membrane removed. The tissue was subsequently permeabilized for 30 min in 0.5% (v/v) Triton X-100/PBS, and blocked for 2 hr in 4% (w/v) BSA/PBS at room temperature. Staining with all primary antibodies was performed overnight at 4°C in the same blocking buffer. For staining with antibodies against GluA2 and $Ca_V1.3$, the protocol was slightly varied: fixation was 20 min at 4°C, followed by decalcification in 10% EDTA (Sigma) for 2 hr; permeabilization and blocking was carried out in a blocking buffer containing 16% donkey serum and 0.3% Triton X-100. After washing, all species-appropriate secondary antibodies (Alexa Fluor 488, 546 or 555, 647: 1:600 dilution in 4% [w/v] BSA/PBS) were applied for 1 hr at room temperature. After further washing, the organs of Corti were mounted, hair bundles facing upward, in ProLong Gold© Antifade (Life Technologies). Samples were imaged using confocal microscopy (LSM700 or LSM880, Zeiss) using 1.4 NA oil immersion objectives for confocal imaging, and using 1.2 NA water immersion objective for Airyscan imaging. The optimal voxel size was adjusted to the green channel (confocal x/y/z: 100/100/350 nm; Airyscan x/y/z: 33/33/222 nm). RIBEYE WT and KO samples were prepared in parallel and images were collected with identical settings.

All images were taken in the apical region, between 200–1000 μm from the apex, of the organ of Corti.

## Image analysis

For three-dimensional analysis of synaptic punctum volume, intensity and colocalization, we used the Imaris software package (Version 8.4, Bitplane). Puncta were initially detected using the spot algorithm allowing for different spot sizes and elongated point spread function in z-direction (initial seed size x-y/z: CtBP1, Ctbp2, Homer, 0.44/1.13 μm; Bassoon 0.37/1.13 μm; GluA3 0.7/1.4 μm; $Ca_V1.3$ 0.56/1.4 μm), followed by co-localizing the spots with a threshold lower than 1. All counts were normalized to the number of IHCs in a region of interest.

To estimate the size of synaptic puncta, we used the surface algorithm in Imaris. For all stainings, the surface detail was set to 0.15 µm, background subtraction to 0.562 µm, and touching object size was 0.750 µm (Bassoon puncta were split at 0.36 µm).

## TEM and tomography

Temporal bones of P21 mice (WT: n = 5, KO: n = 5) were isolated in HBSS, a small hole was introduced at the apex of the cochlea, gently perfused through round and oval windows with freshly prepared 4% (w/v) PFA, 2% (w/v) Glutaraldehyde (Electron Microscopy Sciences) in a buffer containing (in mM) 135 NaCl, 2.4 KCl, 2.4 $CaCl_2$, 4.8 $MgCl_2$, 30 HEPES, pH 7.4 until the fixative became visible at the apex. Temporal bones were soaked for 4 hr in the same buffer and shipped in PBS. Cochleae were microdissected out from fixed adult mouse temporal bones and slowly equilibrated with 30% glycerol as cryoprotection. Tissues were then plunge-frozen in liquid ethane at −180°C using a Leica Biosystems grid plunger. Frozen samples were freeze-substituted using Leica Biosystems AFS in 1.5% uranyl acetate in absolute methanol at −90°C for 2 days, and infiltrated with HM20 Lowicryl resin (Electron Microscopy Sciences) over 2 days at −45°C. Polymerization of the resin was carried out under UV light for 3 days at temperatures between −45°C and 0°C. Ultrathin sections at 100 nm were cut using a Leica ultramicrotome and collected onto hexagonal 300-mesh Cu grids (Electron Microscopy Sciences).

Samples were imaged using a 200kV JEOL 2100 with a Gatan Orius 832 CCD camera. Single images were acquired using DigitalMicrograph (Gatan). Tilt series were acquired using SerialEM (*Mastronarde, 2005*) at 1° tilt increment from −60° to +60°, and tomographic reconstruction of tilt series and segmentation of tomograms was performed with the IMOD software suite (*Kremer et al., 1996*). Fiji software was used to generate projection images from tomograms and for cropping, rotating and adjusting the brightness and contrast of images.

## Hair cell recordings

IHCs from the apical coil of the organ of Corti were dissected from P10–P13 *Ctbp2* littermates of either sex, placed on a plastic coverslip and secured with two strands of dental floss. Cells were visualized with a water immersion lens (100x) attached to a BX51 Olympus upright microscope. Patch-clamp recordings were obtained at room temperature (21–23°C) in an external solution containing in (mM) NaCl 140, KCl 5.4, $CaCl_2$, 2.8, $MgCl_2$ 1, 4-(2-hydroxyethyl)−1-piperazineethanesulfonic acid (HEPES) 10, D-glucose 6, creatine 2, ascorbate 2, sodium pyruvate 2, pH 7.4 adjusted with NaOH. 100 nM apamin was included to block the sK $K^+$ channel. Cesium at 3 mM was used externally to reduce inward leak current. Pipette solution contained (in mM) CsCl 90, TEA, tetraethylammonium chloride 13, 3 NaATP, $MgCl_2$ 4, creatine phosphate 5, ethylene glycol-bis(β-aminoethyl ether)-N,N, N′,N′-tetraacetic acid (EGTA) 1, HEPES 10, pH adjusted with CsOH to 7.2.

## Two-sine method

Hair cells were voltage-clamped with an Axon Multiclamp 700B (Axon Instruments-Molecular Devices, Sunnyvale, CA) and data were recorded at 200–300 kHz with a National Instrument USB-6356 D/A acquisition board and JClamp software (SciSoft). Vesicle fusion was determined by monitoring membrane capacitance as a correlate of surface area change. Capacitance was measured with a dual sinusoidal, FFT-based method (*Santos-Sacchi, 2004*). In this RC analysis method, two voltage frequencies, f1 and f2 (twice the f1 frequency), are summed and the real and imaginary components of the current response are used to determine the magnitudes of the three model components. The time resolution of the Cm measurement is the period of f1, which we varied from 0.16 to 0.32 ms, corresponding to 6250 to 3125 Hz. Stimulation protocols were performed after 10 min in the whole-cell mode to allow $Ca^{2+}$ current run up and full exchange of the internal solution into the cell (*Schnee et al., 2011a*). A dual sinusoid of 40 mV amplitude was delivered superimposed to the desired voltage step. The mean IHC Cm was 10 ± 1 pF N = 25 and the uncompensated series resistance 10 ± 3 MΩ. All voltages were corrected for a 4 mV junction potential.

## Postsynaptic recordings

Whole-cell voltage-clamp recordings were made at terminal endings of SGNs from P17–P21 animals (*Grant et al., 2010*). Cochlea were acutely dissected from the temporal bone in chilled (~4°C)

standard extracellular solution containing 1 mM curare. The apical turn was excised after removing stria vascularis and tectorial membrane, and placed under two dental floss fibers on a recording chamber filled with the standard extracellular solution at room temperature. Recording pipettes were made from 1.5 mm, thick-wall borosilicate glass (WPI). Pipettes were pulled using a multistep horizontal puller (Sutter) and fire polished while applying a pressure to the back end of pipettes to an impedance of 6–8 MΩ (*Johnson et al., 2008*). When whole-cell recording was achieved from a bouton, 5 µM TTX was perfused into the chamber, and EPSCs were recorded in the standard solution (5.8 mM $K^+$) for at least 3 min, then in 40 mM KCl for 1–5 min (depending on the duration of synaptic activity on each nerve). Standard external solution contained (in mM): 5.8 KCl, 140 NaCl, 0.9 $MgCl_2$, 1.3 $CaCl_2$, 0.7 $NaH_2PO_4$, 5.6 D-glucose, 10 HEPES, 2 ascorbate, 2 sodium pyruvate, 2 creatine monohydrate, osmolarity was 305 mosmls, pH was 7.4 (adjusted with NaOH). Intracellular solutions contained (in mM): 135 KCl, 3.5 $MgCl_2$, 0.1 $CaCl_2$, 5.0 EGTA, 5.0 HEPES, 2.5 $Na_2$-ATP, pH = 7.2 (adjusted with KOH), osmolarity ~290 mosmls.

### Data analysis

Data were collected blinded to genotype for the experimentalist. Genotype was revealed during analysis phase to ensure that appropriate n values were obtained between groups. Although gender was noted, no gender differences were observed between groups and so data are presented combined. Data analysis was largely performed using unpaired two-tailed t-tests unless otherwise stated. When non-normal distributions were observed, as in the EPSC histograms, we selected different comparators as described with the data. Similarly, when large n values would bias comparisons, such as when analyzing immunohistochemistry data, we moved to alternative more appropriate tests as described in the text. Data are presented as mean ± standard deviation unless otherwise stated. Sample size is presented in the legends or the text. For physiology experiments, sample size reflects cell numbers which also indicates animal numbers as we did not record from more than one cell in a preparation. For immunohistochemistry, data are presented for animal numbers as well as image, cell and puncta numbers.

The voltage against capacitance plots were arbitrarily fit with an exponential equation of the form: $Y = Y_0 + A_1 \exp^{(-x(x-x_0)/t)}$ where: $Y_0$ is the baseline change in capacitance and is near 0, $A_1$ is the asymptote for the maximal change in capacitance, $x_0$ is the voltage where changes in capacitance measurements were first noted, and t reflects the voltage-dependence of the release ($\Delta Cm/\Delta Vm$). The equation for the Gaussian fit used in *Figure 6E* is: $Y = Y_0 + (A/(w*\sqrt{(\pi/2)}))*\exp^{(-2*((x-x_c)/w)2)}$ where $Y_0$ is baseline, typically 0, A is the area under the curve, w is the width, and $x_c$ is the peak of the Gaussian.

## Acknowledgements

This work was supported by National Institutes of Health, National Institute on Deafness and Other Communication Disorders (NIDCD), Grants R01DC014712 to MAR. MAR was also supported by an International Project Grant from Action on Hearing Loss. Further support came from NIDCD grant F32 4F32 DC013721 grant to MN; Intramural Research Program (Z01-DC000002) and NIDCD Advanced Imaging Core (ZIC DC000081) grants to WS and BK and (RO1DC009913 and core grant P30 44992) to AJR. We thank Autefeh Sajjadi for her help with genotyping animals and basic husbandry. We thank Bill Roberts for comments on the manuscript.

## Additional information

### Funding

| Funder | Grant reference number | Author |
|---|---|---|
| National Institutes of Health | DC009913 | Anthony J Ricci |
| Action on Hearing Loss | | Mark A Rutherford |
| National Institutes of Health | DC014712 | Mark A Rutherford |
| National Institutes of Health | DC013721 | Mamiko Niwa |

| National Institutes of Health | P30 44992 | Anthony J Ricci |
|---|---|---|
| National Institutes of Health | Z01-DC000002 | Willy Sun Bechara Kachar |
| National Institutes of Health | ZIC DC000081 | Willy Sun Bechara Kachar |

The funders had no role in study design, data collection and interpretation, or the decision to submit the work for publication.

## Author contributions

Lars Becker, Data curation, Formal analysis, Validation, Investigation, Visualization, Methodology, Writing—original draft, Writing—review and editing; Michael E Schnee, Conceptualization, Formal analysis, Investigation, Visualization, Writing—original draft, Writing—review and editing; Mamiko Niwa, Software, Formal analysis, Validation, Investigation, Visualization, Writing—original draft, Writing—review and editing; Willy Sun, Data curation, Formal analysis, Investigation, Visualization; Stephan Maxeiner, Resources, Writing—review and editing; Sara Talaei, Data curation, Formal analysis, Writing—review and editing; Bechara Kachar, Data curation, Supervision, Investigation, Visualization, Writing—review and editing; Mark A Rutherford, Resources, Data curation, Formal analysis, Investigation, Writing—original draft, Writing—review and editing; Anthony J Ricci, Conceptualization, Resources, Data curation, Formal analysis, Supervision, Funding acquisition, Investigation, Visualization, Methodology, Writing—original draft, Project administration, Writing—review and editing

## Author ORCIDs

Lars Becker http://orcid.org/0000-0002-3041-3362
Michael E Schnee http://orcid.org/0000-0001-7836-2632
Anthony J Ricci http://orcid.org/0000-0002-1706-8904

## Ethics

Animal experimentation: This study was performed in strict accordance with the recommendations in the Guide for the Care and Use of Laboratory Animals of the National Institutes of Health. All of the animals were handled according to approved institutional animal care and use committee (IACUC) protocol 14345 of Stanford University. The protocol was approved by the Committee on the Ethics of Animal Experiments of the University of Minnesota. All auditory measurements were performed under Ketamine (100mg/kg) and Xylazine,(10mg/kg) and every effort was made to minimize suffering.

## Decision letter and Author response

Decision letter https://doi.org/10.7554/eLife.30241.014
Author response https://doi.org/10.7554/eLife.30241.015

# Additional files

## Supplementary files

• Transparent reporting form
DOI: https://doi.org/10.7554/eLife.30241.012

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
