## [Decision Letter]

Thank you for submitting your article "The presynaptic ribbon facilitates sustained exocytosis at the hair cell afferent fiber synapse" for consideration by *eLife*. Your article has been reviewed by two peer reviewers, and the evaluation has been overseen by a Reviewing Editor and Andrew King as the Senior Editor. The reviewers have opted to remain anonymous.

This decision was reached by discussion between the Reviewing Editor and the reviewers, and the Reviewing Editor has assembled the following comments to help you prepare a revised submission.

Both reviewers highlighted the fact that your study explores a major issue, the role of the ribbon in IHC synaptic exocytosis, and that the article is based on a large, high-quality data set.

However, there is a major issue that needs to be resolved. Both reviewers felt that it was necessary to clarify the discordance between some of the results reported in the two jointly submitted manuscripts, which explore the same mouse Ribeye KO/KO mutant. These discrepancies include, in particular, the modifications to the Ca^2+^ currents of the KO mice. The authors of the two manuscripts are therefore requested to work together to iron out these discrepancies.

The work performed on the zebrafish Ribeye mutants also requires additional attention (Lv C et al., Cell Reports 2016). This work shows that frameshift mutations in the gene encoding Ribeye lead to "ghost ribbons" in zebrafish neuromast hair cells. What evidence is there to suggest that the situation is different in the KO mouse mutant studied? This point should be addressed with care. The results obtained in the zebrafish mutants are generally insufficiently taken into account and discussed in this article, and should be expanded.

*Reviewer #1:*

There are at least 2-3 major discrepancies between this and the companion paper, despite the fact that they are investigating the same RIBEYE KO mouse. First, the ABR thresholds were significantly raised in KO mice in this study, but not in the companion paper. Second, the biophysical properties of the Ca current were identical between the genotypes in this manuscript, but significantly different in the companion paper. This discrepancy could be due to the large variability in the data presented in both manuscripts, possibly indicating some issues with IHC viability. Third, the DCm measurements were found to be significantly different between -43 and -53 mV in the companion paper, whereas this work only shows a difference at -40 mV and not at -45 mV.

Additional points:

1) The authors have highlighted a significant difference in DCm at -40 mV (Figure 6). It would be useful to use this membrane potential to look at the 1st and 2nd components presented in Figure 6. This could show some difference in the immediately releasable pool of vesicles.

2) Figure 7 shows an interesting finding but it is mainly looking at vesicle release from the pool of vesicles further away from the AZs. The main finding of the paper relates to differences in vesicle release from the immediately releasable pool between the two genotypes. Does this pool of vesicles show a different sensitivity to EGTA?

3) The K current recordings in KO mice (Figure 5) do not show the same tail currents as for the WT (Figure 5). Have these recordings been performed using a different protocol?

4) Introduction, second paragraph: I believe that Fuchs 2004 should be Fuchs 2005.

Reviewer #2:

1) One of the most impressive results of the paper is that the amplitude of EPSC distribution displayed a dramatic change becoming double peaked in KO mice (Figure 8). A six-fold increase in small EPSC events was observed, which could perhaps be due to univesicular events. However, the authors do not discuss or emphasize this in the Results or Discussion. These smaller events could be due to single vesicle fusions, which are now more clearly revealed without a synaptic ribbon, whereas in the presence of a synaptic ribbon small single vesicle events are suppressed relative to large multiquantal events. So the ribbon could be promoting large multiquantal events, in this alternative interpretation. Perhaps the authors should also mention this alternative interpretation of the results. I would also recommend more discussion of Figure 8. It's a striking result!

2) The Abstract mentions that: "Surprisingly, multiquantal EPSCs were not smaller in RBE(KO/KO),". I would suggest changing this to: "Surprisingly, large EPSC events were not smaller in RBE(KO/KO), although they become less frequent than in wild type synapses." This seems like a more accurate summary of the results. The authors have not proven that their large EPSCs are multiquantal. They could also be due to smaller amounts of vesicle fusion (or glutamate release) than control, but which nevertheless produce large amplitude EPSCs because of the larger PSDs (or due to higher AMPAR concentrations in the PSDs).

3) Please state the total number of events present in Figure 8. It seems like Figure 8 has a smaller number of events. The CV=0.3 in Figure 8 for control is similar to the CV found by Li et al. (JNeurosci., 2009). Please mention this and also compare with the CV found by Grant et al. (JNeurosci. 2010).

4) "[…]the increase in proportion of EPSCs with lower amplitudes and longer durations is expected[…]" The authors should also mention that larger EPSCs are better phase-locked than smaller EPSCs (see Li et al., Neuron, 2014; see their Figure 8). So one would also expect lower vector strength and reduced phase locking in the KO mice. Indeed, this is exactly what the companion paper of Jean et al. (Moser group) found. So this is well worth mentioning here!

5) Move Figure 5 to the top of Figure 8. It makes more sense to include this high-K depolarization result in Figure 8.

6) In contradiction to this paper, the companion paper of Jean et al. (Moser group) found no change in the ABR thresholds and a slight change in the Ca^2+^ current properties. They also find multiple small conventional active zones per afferent fiber in the RBE-KO-mice. This should be discussed briefly. Maybe the methods are different or the number of animals studied? German mice and American mice should behave the same, right?

---

## [Author Response]

However, there is a major issue that needs to be resolved. Both reviewers felt that it was necessary to clarify the discordance between some of the results reported in the two jointly submitted manuscripts, which explore the same mouse Ribeye KO/KO mutant. These discrepancies include, in particular, the modifications to the Ca^2+^ currents of the KO mice. The authors of the two manuscripts are therefore requested to work together to iron out these discrepancies.

Briefly, our recordings of calcium currents were from animals between the ages of P10-P13, so prehearing but mature in terms of synaptic localization. The recordings were done in whole cell mode. Our data suggests that maturation in terms of localization and voltage dependence were unaffected at this age. Our companion paper used perforated patch measurements at older ages and were done largely in 5 mM external calcium. Thus, the small changes observed between laboratories are likely a function of the technical differences used in making the recordings. We would surmise that the differences between genotype observed by the companion paper are compensatory responses manifesting at older ages or late onset specializations impacted by the absence of the ribbon. Both possibilities are justifiable and we don’t have data supporting either in particular. Importantly though, our data are not in conflict with the companion paper, they represent different time points and thus different conclusions. We have addressed this more carefully in the text.

The work performed on the zebrafish Ribeye mutants also requires additional attention (Lv C et al., Cell Reports 2016). This work shows that frameshift mutations in the gene encoding Ribeye lead to "ghost ribbons" in zebrafish neuromast hair cells. What evidence is there to suggest that the situation is different in the KO mouse mutant studied? This point should be addressed with care. The results obtained in the zebrafish mutants are generally insufficiently taken into account and discussed in this article, and should be expanded.

We have added a paragraph to the Discussion regarding zebrafish data. It is a hard question to answer because neither we, nor our companion paper, nor the original work in retina find any evidence of ghost ribbons and so cannot attest to what differences in response properties between preparations can be ascribed to the ghosts. The morphological differences may be due to the genetic approach not eliminating all ribeye forms in zebrafish. The differences may also be due to strong compensatory mechanisms in zebrafish that are not present in mammal. We looked at CTBP1 as a potential compensatory molecule and found no evidence suggesting that this might be the case. Other data exists suggesting ties between calcium channel currents and ribbon sizes in zebrafish that do not appear to happen in mammal perhaps indicating fundamental differences in plasticity between these two systems. We have elaborated on these points in the Discussion.

Reviewer #1:There are at least 2-3 major discrepancies between this and the companion paper, despite the fact that they are investigating the same RIBEYE KO mouse. First, the ABR thresholds were significantly raised in KO mice in this study, but not in the companion paper. Second, the biophysical properties of the Ca current were identical between the genotypes in this manuscript, but significantly different in the companion paper. This discrepancy could be due to the large variability in the data presented in both manuscripts, possibly indicating some issues with IHC viability. Third, the DCm measurements were found to be significantly different between -43 and -53 mV in the companion paper, whereas this work only shows a difference at -40 mV and not at -45 mV.

We thank the reviewer for their attention to detail and clear comments on this manuscript. We address their three major points here and all others below following the points made. We consider the three points raised as follows:

1) The ABR measurements presented in this manuscript and the companion manuscript agree with each other. The present paper finds a statistically significant difference because we have a larger population of measurements. Our n was selected based on a power analysis using the variance of previous recordings and needing the ability to detect 5-10 dB changes. Thus, we expect that with a larger sample size based on observed variance the companion paper data would also be statistically different. The trend seen in the companion paper is very similar to our data. In the end, there are no methodological or biological differences between the measurements in the two manuscripts.

2) We measured calcium current properties at P10-P12 ages using 1 mM EGTA internal and 2 mM external calcium. The companion paper used perforated patch recordings of older animals and 5 mM calcium. We used earlier ages as a means of assessing currents with a minimum likelihood of accommodation occurring. Any of the differences described above in making the measurements might account for the small differences observed in activation properties between studies. Most likely the companion paper results reflect a later, perhaps activity dependent, secondary response to the loss of the ribbon, but at present neither we nor the companion work have data to support or negate this possibility. Importantly, the data between the manuscripts are not inconsistent with each other and neither support the argument that the ribbon is directly responsible for the clustering of the calcium channels at presynaptic sites.

The question of viability is one that all scientists struggle with, particularly those investigating more mature hair cell function. The criteria used for assessing viability are pretty standard and are based on morphological assessments and electrical properties such as input resistance and membrane capacitance. There is no evidence to suggest that differences exist between data collected between research teams or data sets, in the tissue viability or that cell degradation is impacting the presented data.

3) The take home message from the capacitance measurements is that there is a deficit in release with small stimulations. The two approaches taken for these recordings culminate in this conclusion despite using different technologies. We used whole cell recordings and the two-sine wave technique while the companion work used perforated patch and the single sine technique. The two-sine wave technique benefits from using single stimulations over time to evaluate release changes in real time but suffers from greater noise as there is no averaging between stimuli. We accept this as we have previously demonstrated that multiple stimulations used for averaging can alter the output response. Likely the differences between measurements is time point at which measurements were quantified and potential changes in responses due to different buffering capacities associated with perforated patch vs whole cell recordings. However, we further analyzed our data and include fitting of the capacitance responses at different time points across voltages. We could better compare across genotypes by investigating multiple points across voltages and in doing this found that the sensitivity range was much broader and more overlapping with the companion paper. Thus, in its present form, our manuscript is in even more agreement than in the first version despite a slightly more depolarized sensitivity range. Given the differences in technologies being used, it is remarkable to me that the data are in such good agreement and lead to very similar conclusions.

Additional points:1) The authors have highlighted a significant difference in DCm at -40 mV (Figure 6). It would be useful to use this membrane potential to look at the 1st and 2nd components presented in Figure 6. This could show some difference in the immediately releasable pool of vesicles.

We have modified the figure to include measurements of release rates and levels for both first and second components during smaller depolarizations. We have also pointed out in the text that the superlinear second component does not occur until much later in time with small depolarizations so that the data presented represents largely the first component.

2) Figure 7 shows an interesting finding but it is mainly looking at vesicle release from the pool of vesicles further away from the AZs. The main finding of the paper relates to differences in vesicle release from the immediately releasable pool between the two genotypes. Does this pool of vesicles show a different sensitivity to EGTA?

As we make no assumptions as to the role of the ribbon, we think reporting on all aspects of vesicle release, pools as well as trafficking are relevant. We also agree that looking at individual release components in more detail is valuable and so we have added additional plots to illustrate the effects of buffer on the first component of release. We have done this by studying release at earlier time points and more hyperpolarized potential. Data is added to Figure 7.

3) The K current recordings in KO mice (Figure 5) do not show the same tail currents as for the WT (Figure 5). Have these recordings been performed using a different protocol?

Different tail protocol in initial version, changed figure to provide more comparable stimulation. Basolateral currents show more heterogeneity than might be expected but this was not a function of genotype.

4) Introduction, second paragraph: I believe that Fuchs 2004 should be Fuchs 2005.

Done.

Reviewer #2:1) One of the most impressive results of the paper is that the amplitude of EPSC distribution displayed a dramatic change becoming double peaked in KO mice (Figure 8). A six-fold increase in small EPSC events was observed, which could perhaps be due to univesicular events. However, the authors do not discuss or emphasize this in the Results or Discussion. These smaller events could be due to single vesicle fusions, which are now more clearly revealed without a synaptic ribbon, whereas in the presence of a synaptic ribbon small single vesicle events are suppressed relative to large multiquantal events. So the ribbon could be promoting large multiquantal events, in this alternative interpretation. Perhaps the authors should also mention this alternative interpretation of the results. I would also recommend more discussion of Figure 8. It's a striking result!

We greatly appreciate the enthusiasm of the reviewer and we have expanded to some degree the results and discussion of Figure 8. We are more conservative in our approach to interpreting these data in part because they represent a small component of the total events. Also given the increase in postsynaptic receptors it becomes more difficult to judge changes in amplitudes. By the reviewer’s interpretation one might expect a change in the large amplitude responses, if the ribbon is needed to promote them, but this is not observed, thus we have tempered our Discussion. We see the result as indicating that fewer vesicles available leads to more single events but we found no relationship with amplitude and response timing that might more clearly support this conclusion. We have tried to expound more on these results as suggested.

2) The Abstract mentions that: "Surprisingly, multiquantal EPSCs were not smaller in RBE(KO/KO),". I would suggest changing this to: "Surprisingly, large EPSC events were not smaller in RBE(KO/KO), although they become less frequent than in wild type synapses." This seems like a more accurate summary of the results. The authors have not proven that their large EPSCs are multiquantal. They could also be due to smaller amounts of vesicle fusion (or glutamate release) than control, but which nevertheless produce large amplitude EPSCs because of the larger PSDs (or due to higher AMPAR concentrations in the PSDs).

We have reworded the Abstract as suggested.

3) Please state the total number of events present in Figure 8. It seems like Figure 8 has a smaller number of events. The CV=0.3 in Figure 8 for control is similar to the CV found by Li et al. (JNeurosci., 2009). Please mention this and also compare with the CV found by Grant et al. (JNeurosci. 2010).

We have added the numbers and reference as suggested.

4) "[…]the increase in proportion of EPSCs with lower amplitudes and longer durations is expected[…]" The authors should also mention that larger EPSCs are better phase-locked than smaller EPSCs (see Li et al., Neuron, 2014; see their Figure 8). So one would also expect lower vector strength and reduced phase locking in the KO mice. Indeed, this is exactly what the companion paper of Jean et al. (Moser group) found. So this is well worth mentioning here!

We have added the suggested sentence.

5) Move Figure 5 to the top of Figure 8. It makes more sense to include this high-K depolarization result in Figure 8.

Done.

6) In contradiction to this paper, the companion paper of Jean et al. (Moser group) found no change in the ABR thresholds and a slight change in the Ca^2+^ current properties. They also find multiple small conventional active zones per afferent fiber in the RBE-KO-mice. This should be discussed briefly. Maybe the methods are different or the number of animals studied? German mice and American mice should behave the same, right?

See above comments on ABRs where we largely see the differences being sample size as the companion paper sees the same trend just with more variance.